# Human Neocortical Neurosolver (HNN), a new software tool for interpreting the cellular and network origin of human MEG/EEG data

Samuel A Neymotin[1,2]*, Dylan S Daniels[1], Blake Caldwell[1], Robert A McDougal[3,4], Nicholas T Carnevale[3], Mainak Jas[5,6], Christopher I Moore[1], Michael L Hines[3], Matti Hämäläinen[5,6], Stephanie R Jones[1,7]*

[1]Department Neuroscience, Carney Institute for Brain Sciences, Brown University, Providence, United States; [2]Center for Biomedical Imaging and Neuromodulation, Nathan S. Kline Institute for Psychiatric Research, Orangeburg, United States; [3]Department Neuroscience, Yale University, New Haven, United States; [4]Department of Biostatistics, Yale University, New Haven, United States; [5]Athinoula A. Martinos Center for Biomedical Imaging, Massachusetts General Hospital, Charlestown, United States; [6]Harvard Medical School, Boston, United States; [7]Center for Neurorestoration and Neurotechnology, Providence VAMC, Providence, United States

**\*For correspondence:**
samuel.neymotin@nki.rfmh.org
(SAN);
Stephanie_Jones@brown.edu
(SRJ)

**Competing interests:** The authors declare that no competing interests exist.

**Abstract** Magneto- and electro-encephalography (MEG/EEG) non-invasively record human brain activity with millisecond resolution providing reliable markers of healthy and disease states. Relating these macroscopic signals to underlying cellular- and circuit-level generators is a limitation that constrains using MEG/EEG to reveal novel principles of information processing or to translate findings into new therapies for neuropathology. To address this problem, we built Human Neocortical Neurosolver (HNN, https://hnn.brown.edu) software. HNN has a graphical user interface designed to help researchers and clinicians interpret the neural origins of MEG/EEG. HNN's core is a neocortical circuit model that accounts for biophysical origins of electrical currents generating MEG/EEG. Data can be directly compared to simulated signals and parameters easily manipulated to develop/test hypotheses on a signal's origin. Tutorials teach users to simulate commonly measured signals, including event related potentials and brain rhythms. HNN's ability to associate signals across scales makes it a unique tool for translational neuroscience research.

## Introduction

Modern neuroscience is in the midst of a revolution in understanding the cellular and genetic substrates of healthy brain dynamics and disease due to advances in cellular- and circuit-level approaches in animal models, for example two-photon imaging and optogenetics. However, the translation of new discoveries to human neuroscience is significantly lacking (*Badre et al., 2015*; *Sahin et al., 2018*). To understand human disease, and more generally the human condition, we must study humans. To date, EEG and MEG are the only noninvasive methods to study electrical neural activity in humans with fine temporal resolution. Despite the fact that EEG/MEG provide biomarkers of almost all healthy and abnormal brain dynamics, these so called 'macro-scale' techniques suffer from difficulty in interpretability in terms of the underlying cellular- and circuit-level events. As such, there is a need for a translator that can bridge the 'micro-scale' animal data with the 'macro-

**eLife digest** Neurons carry information in the form of electrical signals. Each of these signals is too weak to detect on its own. But the combined signals from large groups of neurons can be detected using techniques called EEG and MEG. Sensors on or near the scalp detect changes in the electrical activity of groups of neurons from one millisecond to the next. These recordings can also reveal changes in brain activity due to disease.

But how do EEG/MEG signals relate to the activity of neural circuits? While neuroscientists can rarely record electrical activity from inside the human brain, it is much easier to do so in other animals. Computer models can then compare these recordings from animals to the signals in human EEG/MEG to infer how the activity of neural circuits is changing. But building and interpreting these models requires advanced skills in mathematics and programming, which not all researchers possess.

Neymotin et al. have therefore developed a user-friendly software platform that can help translate human EEG/MEG recordings into circuit-level activity. Known as the Human Neocortical Neurosolver, or HNN for short, the open-source tool enables users to develop and test hypotheses on the neural origin of EEG/MEG signals. The model simulates the electrical activity of cells in the outer layers of the human brain, the neocortex. By feeding human EEG/MEG data into the model, researchers can predict patterns of circuit-level activity that might have given rise to the EEG/MEG data. The HNN software includes tutorials and example datasets for commonly measured signals, including brain rhythms. It is free to use and can be installed on all major computer platforms or run online.

HNN will help researchers and clinicians who wish to identify the neural origins of EEG/MEG signals in the healthy or diseased brain. Likewise, it will be useful to researchers studying brain activity in animals, who want to know how their findings might relate to human EEG/MEG signals. As HNN is suitable for users without training in computational neuroscience, it offers an accessible tool for discoveries in translational neuroscience.

---

scale' human recordings in a principled way. This is the ideal problem for computational neural modeling, where the model can have specificity at different scales.

To address this need, we developed the Human Neocortical Neurosolver (HNN), a modeling tool designed to provide researchers and clinicians an easy-to-use software platform to develop and test hypotheses regarding the neural origin of their data. The foundation of the HNN software is a neo-cortical model that accounts for the biophysical origin of macroscale extracranial EEG/MEG recordings with enough detail to translate to the underlying cellular- and network-level activity. HNN's graphical user interface (GUI) provides users with an interactive tool to interpret the neural underpinnings of EEG/MEG data and changes in these signals with behavior or neuropathology.

HNN's underlying model represents a canonical neocortical circuit based on generalizable features of cortical circuitry, with individual pyramidal neurons and interneurons arranged across the cortical layers, and layer-specific input pathways that relay spiking information from other parts of the brain, which are not explicitly modeled. Based on known electromagnetic biophysics underlying macroscale EEG/MEG signals (*Jones, 2015*), the elementary current generators of EEG/MEG (current dipoles) are simulated from the intracellular current flow in the long and spatially-aligned pyramidal neuron dendrites (*Hämäläinen et al., 1993*; *Ikeda et al., 2005*; *Jones, 2015*; *Murakami et al., 2003*; *Murakami and Okada, 2006*; *Okada et al., 1997*). This unique construction produces equal units between the model output and source-localized data (ampere-meters, Am) allowing one-to-one comparison between model and data to guide interpretation.

The extracranial macroscale nature of EEG/MEG limits the space of signals that are typically observed and studied. The majority of studies focus on quantification of event related potentials (ERPs) and low-frequency brain rhythms (<100 Hz), and there are commonalities in these signals across tasks and species (*Buzsáki et al., 2013*; *Shin et al., 2017*). HNN's underlying mathematical model has been successfully applied to interpret the mechanisms and meaning of these common signals, including sensory evoked responses and oscillations in the alpha (7–14 Hz), beta (15–29 Hz) and gamma bands (30–80 Hz) (*Jones et al., 2009*; *Jones et al., 2007*; *Lee and Jones, 2013*;

*Sherman et al., 2016*; *Ziegler et al., 2010*), and changes with perception (*Jones et al., 2007*) and aging (*Ziegler et al., 2010*). The model has also been used to study the impact of non-invasive brain stimulation on circuit dynamics measured with EEG (*Sliva et al., 2018*), and to constrain more reduced 'neural mass models' of laminar activity (*Pinotsis et al., 2017*). In the clinical domain, HNN's model has also been applied to study MEG-measured circuit deficits in autism (*Khan et al., 2015*).

Despite these examples of use, the complexity of the original model and code hindered use by the general community. The innovation in the new HNN software is the construction of an intuitive graphical user interface to interact with the model without any coding. We offer several free and publicly-available resources to assist the broad EEG/MEG community in using the software and applying the model to their studies. These resources include an example workflow, several tutorials (based on the prior studies cited above) to study ERPs and oscillations, and community-sharing resources.

HNN's GUI is designed so that researchers can simultaneously view the model's net current dipole output and microscale features (including layer-specific responses, individual cell spiking activity, and somatic voltages) in both the time and frequency domains. HNN is constructed to be a hypothesis development and testing tool to produce circuit-level predictions that can then be directly tested and informed by invasive recordings and/or other imaging modalities. This level of scalability provides a unique tool for translational neuroscience research.

In this paper, we outline biophysiological and physiological background information that is the basis of the development of HNN, give an overview of tutorials and available data and parameter sets to simulate ERPs and low-frequency oscillations in the alpha, beta, and gamma range, and describe current distribution and online resources (https://hnn.brown.edu). We discuss the differences between HNN and other EEG/MEG modeling software packages, as well as limitations and future directions.

## Results

### Background information on the generation of EEG/MEG signals and uniqueness of HNN

#### Primary currents and the relation to forward and inverse modeling

The concepts of electromagnetic biophysics are succinctly discussed, for example in *Hämäläinen et al. (1993)* and *Hari and Ilmoniemi (1986)*. Here, we briefly review the basic framework of forward and inverse modeling and how they relate to HNN (*Figure 1*).

MEG/EEG signals are created by electrical currents in the brain. The signals are recorded by sensors at least a centimeter from the actual current sources. Given this configuration, a macroscopic scale is employed for the distribution of electrical conductivity. The division between the actual non-ohmic equivalent current sources of activity and the passive ohmic currents is then referred to as primary ($\mathbf{J}^p$) and volume currents ($\mathbf{J}^v$), respectively. As depicted in *Figure 1*, both MEG and EEG are ultimately generated by the primary currents. The primary currents set up a potential distribution (V) that extends through the brain tissue, the cerebrospinal fluid (CSF), the skull, and the scalp, where it is measured as EEG; the passive $\mathbf{J}^v$ is proportional to the electric field (negative gradient of V) and electrical conductivity ($\sigma$). MEG, in general, is generated by both the $\mathbf{J}^p$ and $\mathbf{J}^v$. The total current is the sum of $\mathbf{J}^p$ and $\mathbf{J}^v$: $\mathbf{J} = \mathbf{J}^p + \mathbf{J}^v$, whence $\mathbf{J}^p = \mathbf{J} - \mathbf{J}^v = \mathbf{J} + \sigma\nabla V$. In this definition, the conductivity is considered on a macroscopic scale, omitting the cellular level details. The primary current is the 'battery' of the circuit and is nonzero at the active sites in the brain. Furthermore, the direction of the primary current is determined by the cellular level geometry of the active cells. By locating the primary currents from MEG/EEG we locate the sites of activity.

The task of computing EEG and MEG given $\mathbf{J}^p$ is commonly called forward modeling, and it is governed by Maxwell's equations. However, in the geometry of the head, the integral effect of the volume currents to the magnetic field can be relatively easily taken into account and, therefore, modeling of MEG is in general more straightforward than the precise calculation of the electric potentials measured in EEG. Specifically, a first order approximation, the spherically symmetric conductor model (*Sarvas, 1987*), can often be used (*Tarkiainen et al., 2003*). In this case, all components of the magnetic field can be computed from an analytical formula which is independent of the

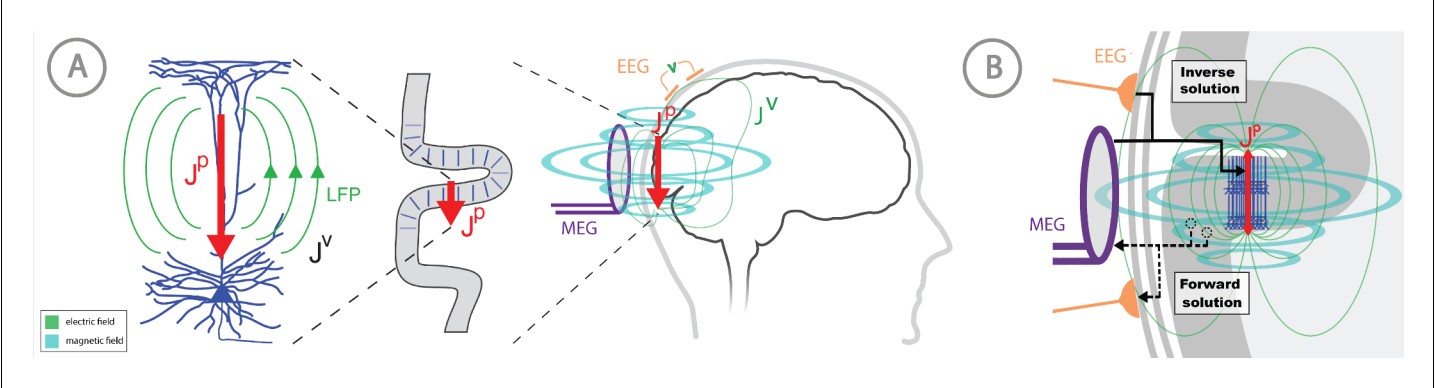

**Figure 1.** Overview of the biophysical origin of MEG/EEG signals and the relationship between HNN and forward/inverse modeling. (**A**) HNN bridges the 'macroscale' extracranial EEG/MEG recordings to the underlying cellular- and circuit-level activity by simulating the primary electrical currents ($J^P$) underlying EEG/MEG, which are generated by the postsynaptic, intracellular current flow in the long and spatially-aligned dendrites of a large population of synchronously-activated pyramidal neurons. (**B**) A zoomed in representation of the relationship between HNN and EEG/MEG forward and inverse modeling. Inverse modeling estimates the location, timecourse and orientation of the primary currents ($J^P$), and HNN simulates the neural activity creating $J^P$, at the microscopic scale. Adapted from *Jones (2015)*.

value of the electrical conductivity as a function of the distance from the center of the sphere (*Sarvas, 1987*). In a more complex case with realistically shaped conductivity compartments, the skull and the scalp can be replaced by a perfect insulator (*Hämäläinen and Sarvas, 1987*; *Hämäläinen and Sarvas, 1989*).

The availability of these forward models opens up the possibility to estimate the locations (**r**) and time course of the primary current activity, $J^P = J^P(\mathbf{r},t)$ from MEG and EEG sensor data, that is inverse modeling, or source estimation. However, this inverse problem is fundamentally ill-posed, and constraints are needed to render the problem unique. The different source estimation methods, such as current dipole fitting, minimum-norm estimates, sparse source estimation methods, and beamformer approaches, differ in their capability to approximate the extent of the source activity and in their localization accuracy; there are presently several open-source software packages for source estimation, for example *Gramfort et al. (2013)*. All of these methods are capable of inferring both the location and direction of the neural currents and their time courses. Importantly, due to physiological considerations, the appropriate elementary primary current source in all of these methods is estimated as a *current dipole,* with units current x distance, that is Am (Ampere x meter).

Thanks to the consistent orientation of the apical dendrites of the pyramidal cells in the cortex this primary current is oriented normal to the cortical mantle and its direction corresponds to the intracellular current flow (*Okada et al., 1997*; *Ikeda et al., 2005*; *Murakami and Okada, 2006*). When source estimation is used in combination with geometrical models of the cortex constructed from anatomical MRI, the current direction can be related to the direction of the outer normal of the cortex: one is thus able to tell whether the estimated current is flowing outwards or inwards at a particular cortical site at a particular point in time. As such, the direction of the current flow can be related to orientation of the pyramidal neuron apical dendrites and inferred as currents flow from soma to apical tuft (up the dendrites) or apical tuft to soma (down the dendrites), see *Figure 1*.

## Inferring the neural origin of the primary currents with HNN

The focus of HNN is to study how $J^P$ is generated by the assembly of neurons in the brain at the microscopic scale. Currently, the process of estimating the primary current sources with inverse methods, or calculating the forward solution from $J^P$ to the measured sensor level signal, is separate from HNN. A future direction is to integrate the top-down source estimation software with our bottom-up HNN model for all-in-one source estimation and circuit interpretation (see Discussion).

HNN's underlying neural model contains elements that can simulate the primary current dipoles ($J^P$) creating EEG/MEG signals in a biophysically principled manner (*Figure 1*). Specifically, HNN simulates the primary current from a canonical model of a layered neocortical column via the net intracellular electrical current flow in the pyramidal neuron dendrites in a direction parallel to the apical

dendrites (see red arrow in *Figures 1* and *2*, and further discussion in Materials and methods) (*Hämäläinen et al., 1993*; *Ikeda et al., 2005*; *Jones, 2015*; *Murakami et al., 2003*; *Murakami and Okada, 2006*; *Okada et al., 1997*). With this construction, the units of measure produced by the model are the same as those estimated from source localization methods, namely, ampere-meters (Am), enabling one-to-one comparison of results. This construction is unique compared to other EEG/MEG modeling software (see Discussion). A necessary step in comparing model results with source-localized signals is an understanding of the direction of the estimated net current in or out of the cortex, which corresponds to current flow down or up the pyramidal neuron dendrites, respectively, as discussed above. Estimation of current flow orientation at any point in time is an option in most inverse solution software that helps guide the neural interpretation, as does prior knowledge of the relay of sensory information in the cortex, see further discussion in the Tutorials part of the Results section.

By keeping model output in close agreement with the data, HNN's underlying model has led to new and generative predictions on the origin of sensory evoked responses and low-frequency rhythms, and on the changes in these signals across experimental conditions (*Jones et al., 2009*; *Jones et al., 2007*; *Khan et al., 2015*; *Lee and Jones, 2013*; *Sherman et al., 2016*; *Sliva et al., 2018*; *Ziegler et al., 2010*) described further below. The macro- to micro-scale nature of the HNN software is designed to develop and test hypotheses that can be directly validated with invasive recordings or other imaging modalities (see further discussion in tutorial on alpha and beta rhythms).

HNN is currently constructed to dissect the cell and network contributions to signals from one source-localized region of interest. Specifically, the HNN GUI is designed to simulate sensory evoked responses and low-frequency brain rhythms from a single region, based on the local network dynamics and the layer-specific thalamo-cortical and cortico-cortical inputs that contribute to the local activity. As such, HNN's underlying neocortical network represents a scalable patch of neocortex containing canonical features of neocortical circuitry (*Figure 2*). Ongoing expansions will include the ability to import other user-defined cell types and circuit models into HNN, simulate LFP and sensor level signals, as well as the the interactions among multiple neocortical areas (see Discussion). Of note, users can still benefit from our software if they are working with data directly from EEG/MEG sensor rather than source-localized signals. The primary currents are the foundation of the sensor signal and, as such, can have similar activity profiles (e.g., compare source-localized tactile evoked response in Figure 4 and sensor-level response in Figure 5).

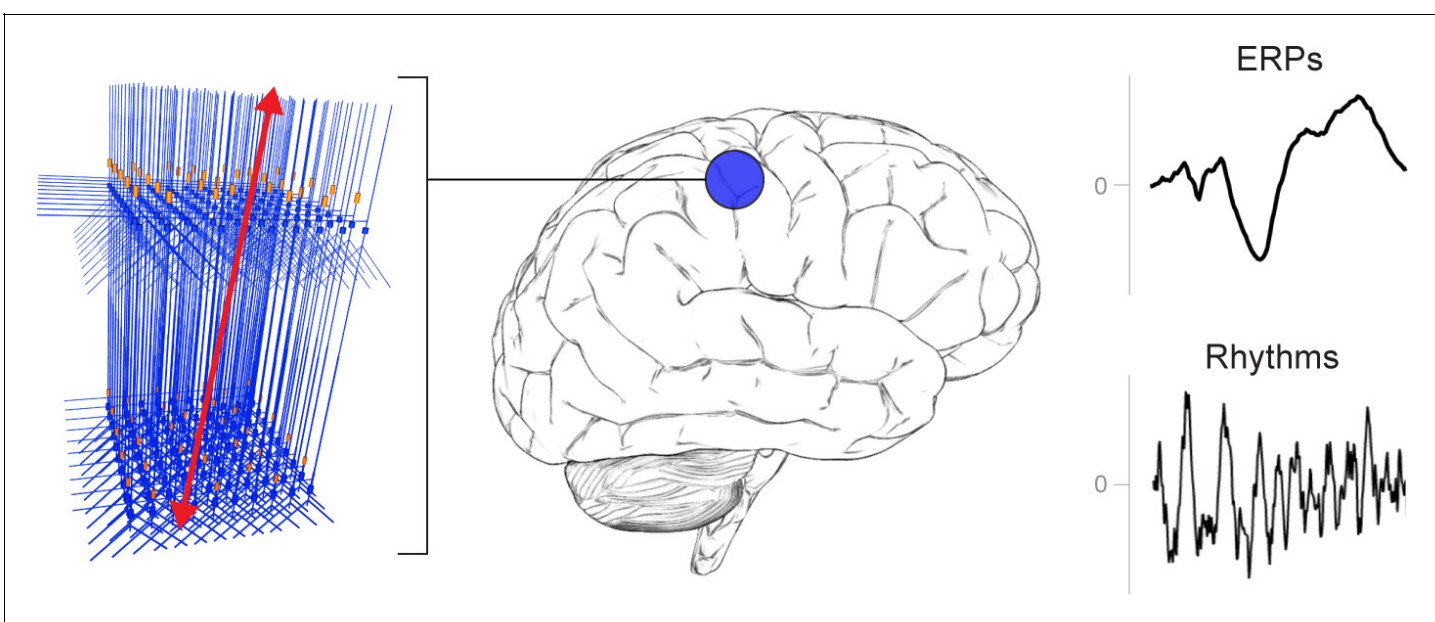

**Figure 2.** A schematic illustration of a canonical patch of neocortex that is represented by HNN's underlying neural model. (Left) 3D visualization of HNN's model (pyramidal neurons drawn in blue, interneurons drawn in orange. (Right) Commonly measured EEG/MEG signals (ERPs and low frequency rhythms) from a single brain area that can be studied with HNN.

# Overview of HNN's default canonical neocortical column template network

## Neocortical column structure

Here, we give an overview of the main features that are important to understand in order to begin exploring the origin of macroscale evoked responses and brain rhythms, and we provide details on how these features are implemented in HNN's template model. Further details can be found in the Materials and methods section, in our prior publications (e.g., *Jones et al., 2009*), and on our website https://hnn.brown.edu.

Given that the primary electrical current that generates EEG/MEG signals comes from synchronous activity in pyramidal neuron (PN) dendrites across a large population, there are several key features of neocortical circuitry that are essential to consider when simulating these currents. While there are known differences in microscale circuitry across cortical areas and species, many features of neocortical circuits are remarkably similar. We assume these conserved features are minimally sufficient to account for the generation of evoked responses and brain rhythms measured with EEG/ MEG, and we have harnessed this generalization into HNN's foundational model, with success in simulating many of these signals using the same template model (see Introduction). These canonical features include:

(I) A 3-layered structure with pyramidal neurons in the supragranular and infragranular layers whose dendrites span across the layers and are synaptically coupled to inhibitory interneurons in a 3-to-1 ratio of pyramidal to inhibitory cells (*Figure 3A*). Of note, cells in the granular layer are not explicitly included in the template circuit. This initial design choice was based on the fact that macroscale current dipoles are dominated by PN activity in supragranular and infragranular layers. Thalamic input to granular layers is presumed to propagate directly to basal and oblique dendrites of PN in the supragranular and infragranular layers. In the model, the thalamic input synapses directly onto these dendrites.

The number of cells in the network is adjustable in the Local Network Parameters window via the Cells tab, while maintaining at 3-to-1 pyramidal to inhibitory interneuron ratio in each layer. The connectivity pattern is fixed, but the synaptic weights between cell types can be adjusted in the Local Network menu and the Synaptic Gains menu. Macroscale EEG/MEG signals are generated by the synchronous activity in large populations of PN neurons. Evoked responses are typically on the order

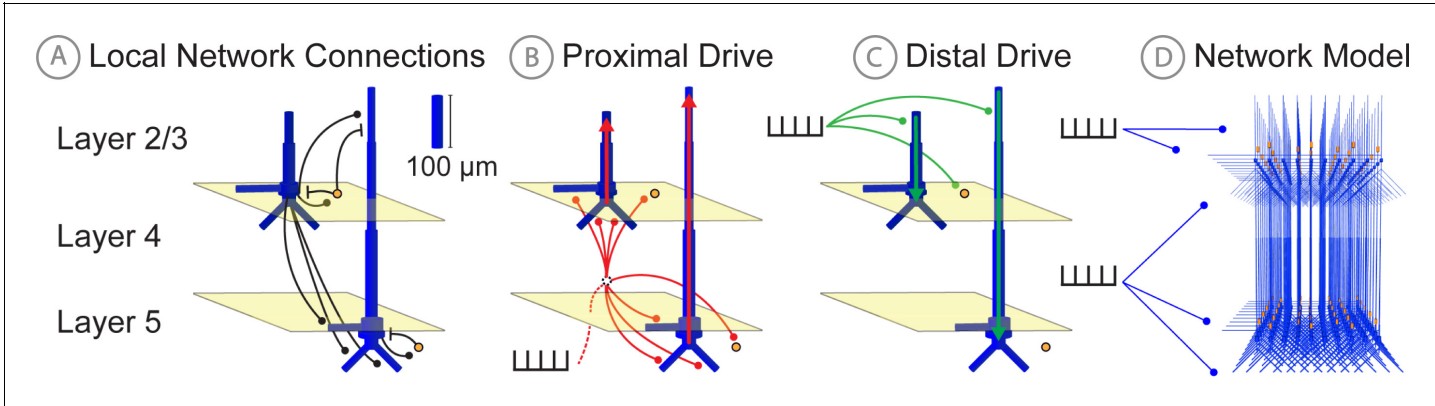

**Figure 3.** Schematic illustrations of HNN's underlying neocortical network model. (**A**) Local Network Connectivity: GABAergic (GABAA/GABAB; lines) and glutamatergic (AMPA/NMDA; circles) synaptic connectivity between single-compartment inhibitory neurons (orange circles) and multi-compartment layer 2/3 and layer five pyramidal neurons (blue neurons). Excitatory to excitatory connections not shown, see Materials and methods. (**B**) Exogenous proximal drive representing lemniscal thalamic drive to cortex. User defined trains or bursts of action potentials (see tutorials described in text) are simulated and activate post-synaptic excitatory synapses on the basal and oblique dendrites of layer 2/3 and layer five pyramidal neurons as well as the somata of layer 2/3 and layer five interneurons. These excitatory synaptic inputs drive current flow up the dendrites towards supragranular layers (red arrows). (**C**) Exogenous distal drive representing cortical-cortical inputs or non-lemniscal thalamic drive that synapses directly into the supragranular layer. User defined trains of action potentials are simulated and activate post-synaptic excitatory synapses on the distal apical dendrites of layer 5 and layer 2/3 pyramidal neurons as well as the somata of layer 2/3 interneurons. These excitatory synaptic inputs push the current flow down towards the infragranular layers (green arrows). (**D**) The full network contains a scalable number of pyramidal neurons in layer 2/3 and layer 5 in a 3-to-1 ratio with inhibitory interneurons, activated by user defined layer specific proximal and distal drive (see Materials and methods for full details).

of 10 – 100nAm, and are estimated to be generated by the synchronous spiking activity of the order of tens of thousands of pyramidal neurons. Low-frequency oscillations are larger in magnitude and are on the order of 100–1000 nAm, and are estimated to be generated by the subthreshold activity of on the order of a million pyramidal neurons (*Jones et al., 2009*; *Jones et al., 2007*; *Murakami and Okada, 2006*). While HNN is constructed with the ability to adjust local network size, the magnitude of these signals can also be conveniently matched by applying a scaling factor to the model output, providing an estimate of the number of neurons that contributed to the signal.

(II) Exogenous driving input through two known layer-specific pathways. One type of input represents excitatory synaptic drive that comes from the lemniscal thalamus and contacts the cortex in the granular layers, which then propagates to the proximal PN dendrites in the supragranular and infragranular layers and somata of the inhibitory neurons; this input is referred to as proximal drive (*Figure 3B*). The other input represents excitatory synaptic drive from higher-order cortex or non-specific thalamic nuclei that synapses directly into the supragranular layers and contacts the distal PN dendrites and somata of the inhibitory neuron; this input is referred to as distal drive (*Figure 3C*). The networks that provide proximal and distal input to the local circuit (e.g., thalamus and higher order cortex) are not explicitly modelled, but rather these inputs are represented by simulated trains of action potentials that activate excitatory post-synaptic receptors in the local network. The temporal profile of these action potentials is adjustable depending on the simulation experiment and can be represented as single spikes, bursts of input, or rhythmic bursts of input. There are several ways to change the pattern of action potential drive through different buttons built into the HNN GUI: Evoked Inputs, Rhythmic Proximal Inputs, and Rhythmic Distal Inputs. The dialog boxes that open with these buttons allow creation and adjustment of patterns of evoked response drive or rhythmic drive to the network (see tutorials described in Results section for further details).

(III) Exogenous drive to the network can also be generated as excitatory synaptic drives following a Poisson process to the somata of chosen cell classes or as tonic input simulated as a somatic current clamp with a fixed current injection. The timing and duration of these drives is adjustable.

Further details of the biophysics and morphology of the cells and of the architecture of the local synaptic connectivity profiles in the template network can be found in the Materials and methods section. As the use of our software grows, we anticipate other cells and network configurations will be made available as template models to work with via open source sharing (see Discussion).

## Parameter tuning in HNN's template network model

HNN's template model is a large-scale model simulated with thousands of differential equations and parameters, making the parameter optimization process challenging. The process for tuning this canonical model and constraining the space of parameters to investigate the origin of ERPs and low-frequency oscillations was as follows. First, the individual cell morphologies and physiologies were constrained so individual cells produced realistic spiking patterns to somatic injected current (detailed in Materials and methods). Second, the local connectivity within and among cortical layers was constructed based on a large body of literature from animal studies (detailed in Materials and methods). All of these equations and parameters were then fixed, and the only parameters that were originally tuned to simulate ERPs and oscillations were the timing and the strength of the exogenous drive to the local network. This drive represented our 'simulation experiment' and was based on our hypotheses on the origin of these signals motivated by literature and on matching model output to features of the data (see tutorials described in Results). The HNN GUI was constructed assuming ERPs and low-frequency oscillations depend on layer-specific exogenous drives to the network. The simulation experiment workflow and tutorials described below are in large part based on 'activating the network' by defining the characteristics of this layer-specific drive. Default parameter sets are provided as a starting point from which the underlying parameters can be inter-actively manipulated using the GUI, and additional exogenous driving inputs can be created or removed.

Automated parameter optimization is also available in HNN and is specifically designed to accurately reproduce features of an ERP waveform based on the temporal spacing and strength of the exogenous driving inputs assumed to generate the ERP. Before taking advantage of HNN's automated parameter optimization, we strongly encourage users to begin by understanding our ERP tutorial and by hand-tuning parameters using one of our default parameter sets to get an initial

representation of the recorded data. The identification of an appropriate number of driving inputs and their approximate timings and strengths serves as a starting point for the optimization procedure (described in the ERP Model Optimization section below). Hand tuning of parameters and visualizing the resultant changes in the GUI will enable users to understand how specific parameter changes impact features of the current dipole waveform.

Importantly, the biophysical constraints on the origin of the current dipoles signal (discussed above) will dictate the output of the model and necessarily limit the space of parameter adjustments that can accurately account for the recorded data. The same principle underlies the fact that a limited space of signals are typically studied at the macroscale (ERPs and low-frequency oscillations). A parameter sensitivity analysis on perturbations around the default ERP parameter sets confirmed that a subset of the parameters have the strongest influence on features of the ERP waveform (see Supplementary Materials). Insights from GUI-interactive hand tuning and sensitivity analyses can help narrow the number of parameters to include in the subsequent optimization procedure and greatly decrease the number of simulations required for optimization.

## HNN GUI overview and interactive simulation experiment workflow

The HNN GUI is designed to allow researchers to link macro-scale EEG/MEG recordings to the underlying cellular- and network-level generators. Currently available visualizations include a direct comparison of simulated electrical sources to recorded data with calculated goodness of fit estimates, layer-specific current dipole activity, individual cell spiking activity, and individual cell somatic voltages (*Figure 4B–D*). Results can be visualized in both the time and frequency domain. Based on its biophysically detailed design, the output of HNN's model and recorded source-localized data have the same units of measure (Am). By closely matching the output of the model to recorded data in an interactive manner, users can test and develop hypotheses on the cell and network origin of their signals.

The process for simulating evoked responses or brain rhythms from a single region of interest is to first define the network structure and then to 'activate' the network with exogenous driving input based on your hypotheses and simulation experiment. HNN's template model provides the initial network structure. The choice of 'activation' to the network depends on the simulation experiment. The GUI design is motivated by our prior published studies and was built specifically to simulate sensory evoked responses, spontaneous rhythms, or a combination of the two (*Jones et al., 2009*; *Jones et al., 2007*; *Khan et al., 2015*; *Lee and Jones, 2013*; *Sherman et al., 2016*; *Sliva et al., 2018*; *Ziegler et al., 2010*). The tutorials described in the Results section below detail examples of how to 'activate' the network to simulate sensory evoked responses and spontaneous rhythms. Here, we outline a typical simulation experiment workflow.

In practice, users apply the following interactive workflow, as in *Figure 4* and detailed further in the tutorials with an example tactile evoked response from somatosensory cortex (data from *Jones et al., 2007*).

(Step 1) Load EEG/MEG data (*blue*). (Step one is optional.)

(Step 2) Define the cortical column network structure. The default template network is automatically loaded when HNN starts. Default parameters describing the local network can be adjusted by clicking the Set Parameters button on the GUI and then Local Network Parameters, or directly from the Local Network Parameters button on the GUI (*Figure 4A*).

(Step 3) 'Activate' the local network by defining layer-specific, exogenous driving inputs (*Figure 3B,C*). The drive represents input to the local circuit from thalamus and/or other cortical areas and can be in the form of (i) spike trains (single spikes or bursts of rhythmic input) that activate post-synaptic targets in the local network, (ii) current clamps (tonic drive), or (iii) noisy (Poisson) synaptic drive. The choice of input parameters depends on your hypotheses and 'simulation experiment'. In the example simulation, predefined evoked response parameters were loaded in via the Set Parameters From File button and choosing the file 'ERPYes100Trials.param'; this is also the default evoked response parameter set loaded when starting HNN (*Figure 4B*). The Evoked Input parameters are then viewed in the Set Parameters dialog box under Evoked Inputs (*Figure 4C*). The Evoked Inputs parameters are described further in the tutorials below.

(Step 4) Run simulation and directly compare model output (*black*) and data (*purple*) with goodness of fit calculations (root mean squared error, RMSE, between data and averaged simulation) (*Figure 4D*).

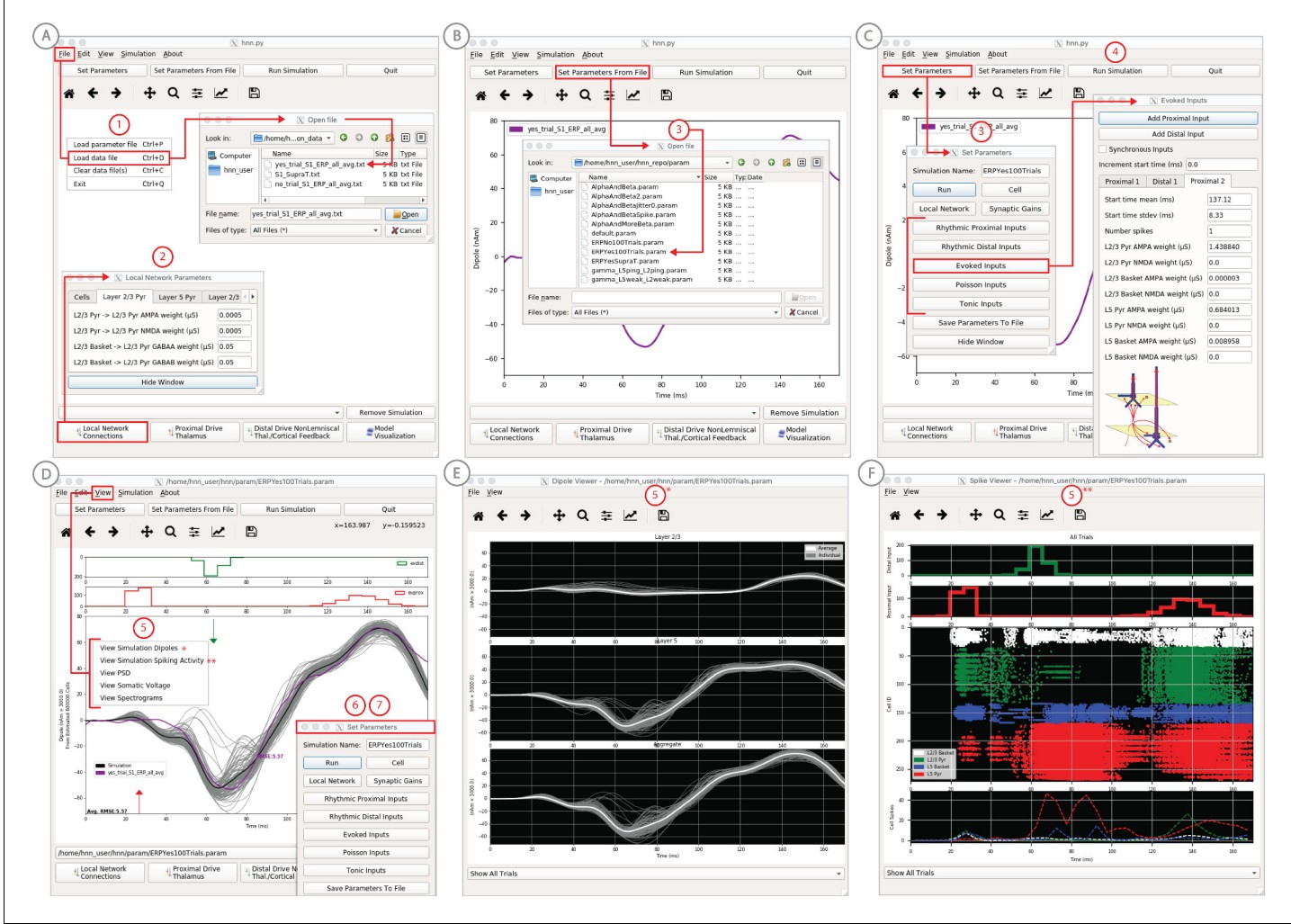

**Figure 4.** An example workflow showing how HNN can be used to link the macroscale current dipole signal to the underlying cell and circuit activity. The example shown is for a perceptual threshold level tactile evoked response (50% detected) from SI (*Jones et al., 2007*; see ERP Tutorial text for details). (**A**) Steps 1 and 2: load data and define the local network structure. (**B**) Step 3: activate the local network, starting with a predefined parameter set; shown here for the parameter set for perceptual threshold-level evoked response (ERPYes100Trials.param) (**C**) Step 3 and 4: adjust the evoked input parameters according to user defined hypotheses and simulation experiment, and run the simulation. (**D**) Step 5: visualize model output; the net current dipole will be displayed in the main GUI window and microcircuit details, including layer-specific responses, cell membrane voltages, and spiking profiles (**E and F**) are shown by choosing them from the View pull down menu. Parameters can be adjusted to hypothesized circuit changes under different experimental conditions (e.g. see *Figure 5*).

The online version of this article includes the following source data and figure supplement(s) for figure 4:

**Source data 1.** Average S1 ERP from detected threshold-level stimulus.
**Figure supplement 1.** Sensitivity analysis results of the perceptual threshold level evoked response example showing the relative contribution of each input's parameters on variance.

(Step 5) Visualize microcircuit details, including layer-specific responses, cell membrane voltages, and spiking profiles by choosing from the View pull down menu (*Figure 4D,E,F*).

(Step 6) Adjust parameters through the Set Parameters dialog box to develop and test predictions on the circuit mechanisms that provide the best fit to the data. With any parameter adjustment, the change in the dipole signal can be viewed and compared with the prior simulation to infer how specific parameters impact the current dipole waveform. Prior simulations can be maintained in the GUI or removed. For ERPs, automatic parameter optimization can be iteratively applied to tune the parameters of the exogenous driving inputs to find those that provide the best initial fit between the simulated dipole waveform and the EEG/MEG data (see further details below).

(Step 7) To infer circuit differences across experimental conditions, once a fit to one condition is found, adjustments to relevant cell and network parameters can be made (guided by user-defined hypotheses), and the simulation can be re-run to see if predicted changes account for the observed differences in the data A list of the GUI-adjustable parameters in the model can be found in the 'Tour of the GUI' section of the tutorials on our website. HNN's GUI was designed so that users could easily find the adjustable parameters from buttons and pull down menus on the main GUI leading to dialogue boxes with explanatory labels.

As a specific example on how to use HNN as a hypothesis testing tool, we have used HNN to evaluate hypothesized changes in EEG-measured neural circuit dynamics with non-invasive brain stimulation (*Figure 5*). We measured somatosensory evoked responses from brief threshold-level taps to the middle finger tip before and after 10 min of ~10 Hz transcranial alternating current stimulation (tACS) over contralateral somatosensory cortex (see *Sliva et al., 2018* for details). The magnitude of an early peak near ~70 ms in the tactile evoked response increased after the tACS session (*Figure 5*, top left). Based on prior literature, we hypothesized that the observed difference was due to changes in synaptic efficacy in the local network induced by the tACS (*Kronberg et al., 2017*; *Rahman et al., 2017*). To test this hypothesis, we first used HNN to simulate the pre-tACS evoked responses, following the evoked response tutorial in our software (see Tutorial below). Once the pre-tACS condition was accounted for, we then adjusted the synaptic gain between the excitatory and inhibitory cells in the network using the HNN GUI and re-simulated the tactile evoked responses. We tested several possible gain changes between the populations. HNN showed that a two-fold increase in synaptic strength of the inhibitory connections, as opposed to an increase in the excitatory connections or in total synaptic efficacy, could best account for the observed differences in the data (compare blue in red curves in *Figure 5*). By viewing the cell spiking profiles in each condition (*Figure 5*, bottom right), HNN further predicted that the increase in the magnitude of the ~70 ms peak coincided with increased firing in the inhibitory neuron population and decreased firing in the excitatory pyramidal neurons in the post-tACS compared to the pre-tACS window. These detailed predictions can guide further experiments and follow-up testing in animal models or with other human imaging experiments. Follow up testing of model derived predictions is described further in the alpha/beta tutorial below.

## Tutorials on ERPs and low-frequency oscillations

HNN's tutorials are designed to teach users how to simulate the most commonly studied EEG/MEG signals, including sensory evoked responses and low-frequency oscillations (alpha, beta, and gamma rhythms) by walking users through the workflow we applied in our prior studies of these signals. The data and parameter sets used in these studies are distributed with the software, and the interactive GUI design was motivated by this workflow. In completing each tutorial, users will have a sense of the basic structure of the GUI and the process for manipulating relevant parameters and viewing results. From there, users can begin to develop and test hypotheses on the origin of their own data. Below we give a basic overview of each tutorial. The HNN website (https://hnn.brown.edu) provides additional information and example exercises for further exploration.

### Sensory evoked responses

We have applied HNN to study the neural origin of tactile evoked responses localized with inverse methods to primary somatosensory cortex from MEG data (*Jones et al., 2007*). In this study, the tactile evoked response was elicited from a brief perceptual threshold level tap - stimulus strength maintained at 50% detection - to the contralateral middle finger tip during a tactile detection experiment (experimental details in *Jones et al., 2009* and *Jones et al., 2007*). The average tactile evoked response during detected trials is shown in *Figure 4*. The data from this study is distributed with HNN installation.

Following the workflow described above, the process for reproducing these results in HNN is as follows.

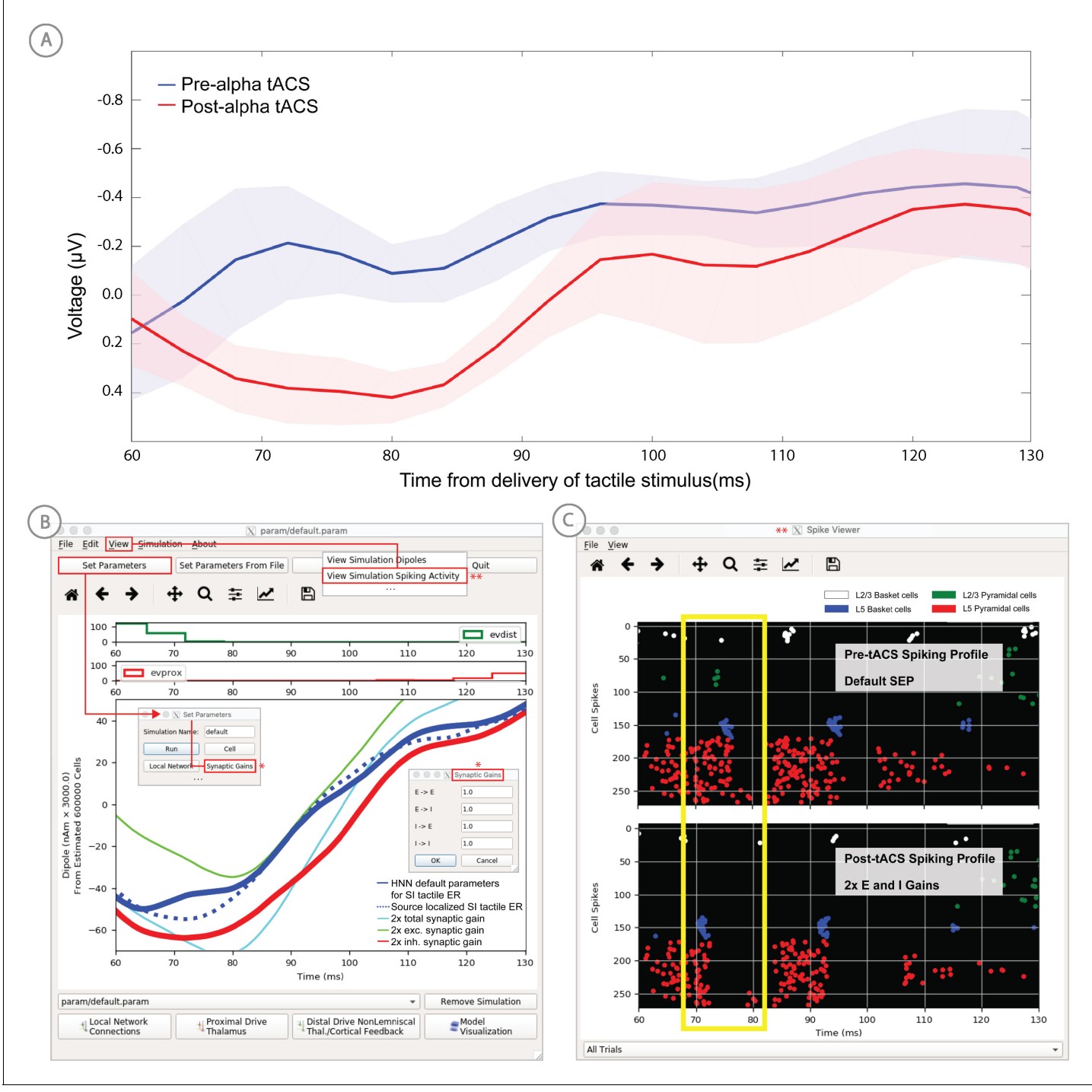

**Figure 5.** Application of HNN to test alternative hypotheses on the circuit level impact of tACS on the somatosensory tactile evoked response (adapted from *Sliva et al., 2018*). (**A**) The early tactile evoked response from above somatosensory cortex before and after 10 min of 10 Hz alternating current stimulation over SI shows that the ~70 ms peak is more prominent in the post-tACS condition. Note that the timing of this peak in the sensor level signal is analogous to the 70 ms peak in the source localized signal in *Figure 4B*, since the tactile stimulation was the same in both studies and the early signal from SI is similar both at the source and sensor level. (**B**) HNN was applied to investigate the impact of several possible tACS induced changes in local synaptic efficacy and identify which could account for the observed evoked response data. The parameters in HNN were first adjusted to account for the pre-tACS response using the default HNN parameter set (solid blue line). The synaptic gains between the different cell types was then adjusted through the Set Parameters dialog box to predict that 2x gain in the local inhibitory synaptic weights best accounted for the post-tACS evoked response. (**C**) Simultaneous viewing of the cell spiking activity further predicted that there is less pyramidal neuron spiking at 70 ms post-tACS, despite the more prominent 70 ms current dipole peak.

## Steps 1 and 2

Load the evoked response data distributed with HNN, 'yes_trial_SI_ERP_all_avg.txt'. The data shown in *Figure 4B* will be displayed. Adjust parameters defining the automatically loaded default local network, if desired.

## Step 3

'Activate' the local network. In prior publications, we showed that this tactile evoked response could be reproduced in HNN by 'activating' the network with a sequence of layer-specific proximal and distal spike train drive to the local network, which is distributed with HNN in the file 'ERPYes100-Trials.param'.

The sequence described below was motivated by intracranial recordings in non-human primates, which guided the initial hypothesis testing in the model. Additionally, we established with inverse methods that at the prominent ~70 ms negative peak (*Figure 4D*), the orientation of the current was into the cortex (e.g., down the pyramidal neuron dendrites), consistent with prior intracranial recordings (see *Jones et al., 2007*). As such, in this example, negative current dipole values correspond to current flow down the dendrites, and positive values up the dendrites. In sensory cortex, the earliest evoked response peak corresponds to excitatory synaptic input from the lemniscal thalamus that leads to current flow out of the cortex (e.g., up the dendrites). This earliest evoked response in somatosensory cortex occurs at ~25 ms. The corresponding current dipole positive peak is small for the threshold tactile response in *Figure 4D*, but clearly visible in Figure 11 for a suprathreshold (100% detection) level tactile response.

The drive sequence that accurately reproduced the tactile evoked response consisted of 'feedforward'/proximal input at ~25 ms post stimulus, followed by 'feedback'/distal input at ~60 ms, followed by a subsequent 'feedforward'/proximal input at ~125 ms (Gaussian distribution of input times on each simulated trial, *Figure 4C*). This 'activation' of the network generated spiking activity and a pattern of intracellular dendritic current flow in the pyramidal neuron dendrites in the local network to reproduce the current dipole waveform, many features of which fell naturally out of the local network dynamics (details in *Jones et al., 2007*). This sequence can be interpreted as initial 'feedforward' input from the lemniscal thalamus followed by 'feedback' input from higher-order cortex or non-lemniscal thalamus, followed by a re-emergent leminscal thalamic drive. A similar sequence of information flow likely applies to most sensory evoked signals. The inputs are distinguished with red and green arrows (corresponding to proximal and distal input, respectively) in the main GUI window. The number, timing, and strength (post-synaptic conductance) of the driving spikes were manually adjusted in the model until a close representation of the data was found (see section on parameter tuning above). To account for some variability across trials, the exact time of the driving spikes for each input was chosen from a Gaussian distribution with a mean and standard deviation (see Evoked Inputs dialog box, *Figure 4C*, and green and red histograms on the top of the GUI in *Figure 4D*). The gray curves in *Figure 4D* show 25 trials of the simulation (decreased from 100 trials in the Set Parameters, Run dialog box) and the black curve is the average across simulations. The top of the GUI windows displays histograms of the temporal profile of the spiking activity providing the sequence of proximal (red) and distal (green) synaptic input to the local network across the 25 trials. Note, a scaling factor was applied to net dipole output to match to the magnitude of the recorded ERP data and used to predict the number of neurons contributing to the recorded ERP. This scaling factor is chosen from Set Parameters, Run dialog box, and is shown as 3000 on the y-axis of the main GUI window in *Figure 4D*. Note that the scaling factor is used to predict the number of pyramidal neurons contributing to the observed signal. In this case, since there are 100 pyramidal neurons in each of layers 2/3 and 5, that amounts to 600,000 neurons (200 neurons x 3000 scaling factor) contributing to the evoked response, consistent with the experimental literature (described in *Jones et al., 2009* and *Jones et al., 2007*).

Based on the assumption that sensory evoked responses will be generated by a layer-specific sequence of drive to the local network similar to that described above, HNN's GUI was designed for users to begin simulating evoked responses by starting with the aforementioned default sequence of drive that is defined when starting HNN and by loading in the parameter set from the 'ERPYes100Trials.param' file, as described above. The Evoked Inputs dialog box (*Figure 4C*) shows the parameters of the proximal and distal drive (number, timing, and strength) used to produce the

evoked response in *Figure 4D*. Here, there were two proximal drives and one distal drive to the network. These parameters were found by first hand tuning the inputs to get a close representation of the data and then running the parameter optimization procedure described below.

## Step 4

The evoked response shown in *Figure 4* is reproduced by clicking the 'Run Simulation' button at the top of the GUI, and the RMSE of the goodness of fit to the data is automatically calculated and displayed. Additional network features can also be visualized through pull down menus (Step 5).

Evoked response parameters can now be adjusted, and additional inputs can be created or removed to account for the user-defined 'simulation experiment' and hypothesis testing goals (Step 6). With each parameter change, a new parameter file will be saved by renaming the simulation under 'Simulation Name' in the 'Set Parameters' dialog box (see *Figure 4C*). From here, other cell or network parameters can be adjusted to compare across conditions (Step 7).

## Alpha and beta rhythms

We have applied HNN to study the neural origin of spontaneous rhythms localized to the primary somatosensory cortex from MEG data; it is often referred to as the mu-rhythm, and it contains a complex of (7–14 Hz) alpha and (15–29 Hz) beta frequency components (*Jones et al., 2009*). A 1 s time frequency spectrogram of the spontaneous unaveraged SI rhythm from this study is shown in *Figure 6a*. This data is distributed on the HNN website ('SI_ongoing.txt'), and contains 1000 1 s epochs of spontaneous data (100 trials each from 10 subjects). The data is plotted in HNN through the 'View → View Spectrograms' menu item, followed by 'Load Data' and then selecting the 'SI_ongoing.txt' file. Note that it may take a few minutes to calculate the wavelet transforms for all 1000 1 s trials included. Next, select an individual trial (e.g. trial 32) from the drop-down menu. The dipole waveform from a single 1 s epoch will then be shown in the top.

The corresponding time-frequency spectrogram is automatically calculated and displayed at the bottom, as seen in *Figure 6A*. The default colormap indicating spectral power is the 'jet' scheme with blue representing low power and red representing high power. This is the same colormap used in the spectrograms of prior publications using the HNN model for studying low-frequency

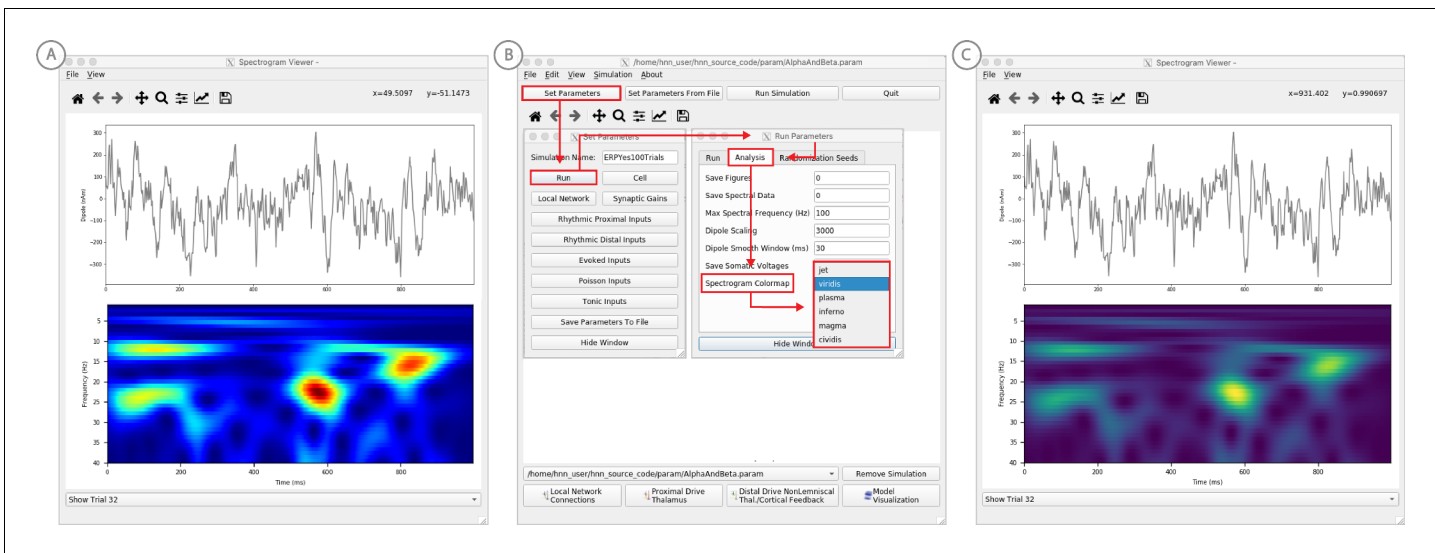

**Figure 6.** Example spontaneous data from a current dipole source in SI showing transient alpha (~7–14 Hz Hz) and beta (~15–29 Hz) components (data as in *Jones et al., 2009*). The data file ('SI_ongoing.txt') used to generate these outputs is provided with HNN and plotted through the 'View → View Spectrograms' menu item, followed by 'Load Data', and then selecting the file. (**A**) The spectrogram viewer with the default 'jet' colormap. (**B**) The configuration option to change the colormap to other perceptually uniform colormaps, where the lightness value increases monotonically. (**C**) The spectrogram viewer updated with the perceptually uniform 'viridis' colormap.

The online version of this article includes the following source data for figure 6:

**Source data 1.** S1 pre-stimulus activity.

oscillations. However, HNN allows the user to plot the spectrogram using different standardized colormaps that are perceptually uniform, meaning they have a lightness value that increases monotonically (*Pauli, 1976*). The configuration option for changing the colormap and the colormap options are shown in *Figure 6B*. The spectrogram resulting from choosing the perceptually uniform 'viridis' colormap is shown in *Figure 7C*. Note that updating the spectrogram colormap requires repeating the 'View → View Spectrograms' menu selection.

Notice that this rhythm contains brief bouts of alpha or beta activity that will occur at different times in different trials due to the spontaneous, non-stationary nature of the signals. Such non-time-locked oscillations are often referred to as spontaneous and/or 'induced' rhythms. When the non-negative spectrograms are averaged across trials, the intermittent bouts of high power alpha and beta activity accumulate without cancellation, and bands of alpha and beta activity appear continuous in the spectrogram (data not shown, see *Jones, 2016*; *Jones et al., 2007*) and will create peaks at alpha and beta in a power spectral density (*Figure 8C*). Since the alpha and beta components of this rhythm are not time locked across trials, it is difficult to directly compare the waveform of the recorded data with model output. Rather, to assess the goodness of fit of the model, we compared features of the simulated rhythm to the data (see *Jones et al., 2009*), including peaks in the power spectral density, as described below. Since we can not directly compare the waveform of this rhythm with the model output, rather than first loading the data, we begin this tutorial with Step 3, 'activating' the network, using the default local network defined when starting HNN.

## Step 3

'Activate' the local network. In prior publications, we have simulated non-time-locked spontaneous low-frequency alpha and beta rhythms through patterns of rhythmic drive (repeated bursts of spikes) through proximal and distal projection pathways. These patterns of drive were again motivated by literature and by tuning the parameters to match features of the model output to the recorded data (see *Jones et al., 2009*; *Sherman et al., 2016*).

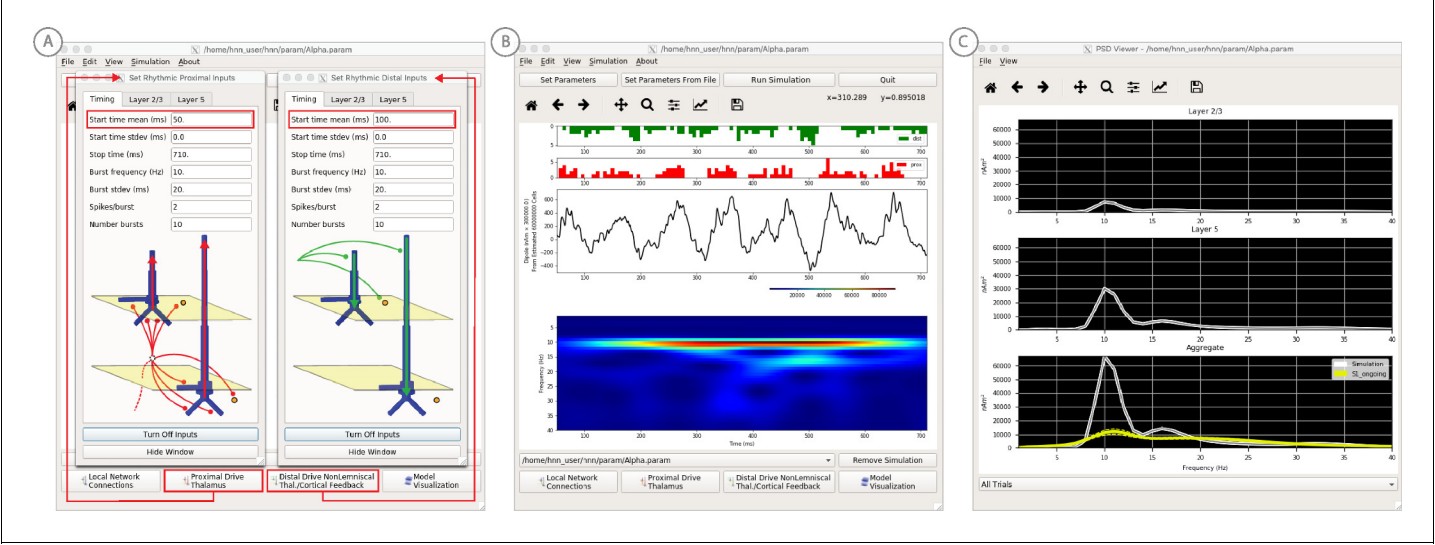

**Figure 7.** An example workflow for simulating alpha frequency rhythm (*Jones et al., 2009*; *Ziegler et al., 2010*; see Alpha and Beta Rhythms Tutorial text for details). (**A**) Here we are using the default HNN network configuration and not directly comparing the waveform to data, so begin with Step 3: activate the local network. Motivated by prior studies (see text), in this example alpha rhythms were simulated by driving the network with ~10 Hz bursts (presumed to be generated by thalamus) to the local network through proximal and distal projection pathways. The parameter set describing these burst is provided in the Alpha.param file and loaded through the Set Parameters From File button. Adjustable burst drive parameter are shown and here were set with a 50 ms delay between the ~10 proximal and distal drive (red boxes). (**B**) Step 4: running the simulation with the 'Run Simulation' button, shows that a continuous alpha rhythm emerged in the current dipole signal (middle dipole time trace; bottom time-frequency representation). Green and red histograms at the top display the defined distal and proximal burst drive patterns, respectively. (**C**) Step 5: additional network features, including layer specific power spectral density plots as shown can be visualized through the 'View' pull down menu, and compared to data (here compared to the spontaneous SI data shown in *Figure 6A*). Features of the burst drive can be adjusted (panel A) and corresponding changes in the current dipole signals studied (Steps 6 and 7, see *Figure 8*).

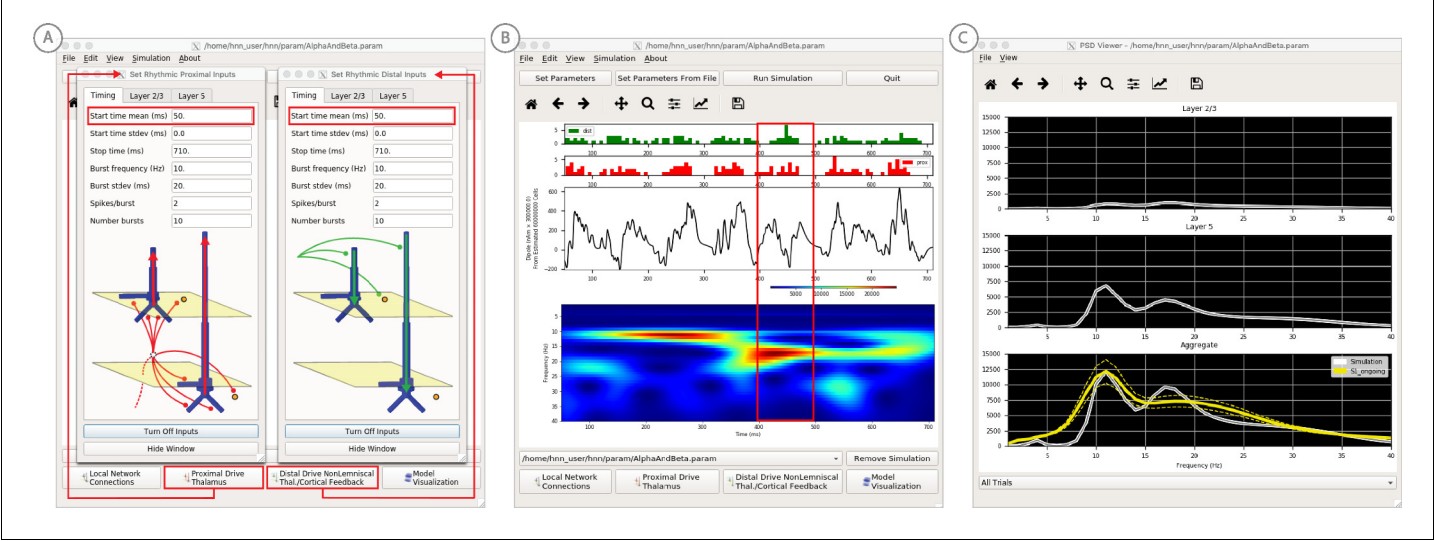

**Figure 8.** An example workflow for simulating transient alpha and beta frequency rhythm as in the spontaneous SI rhythms shown in *Figure 6A* (*Jones et al., 2009*; *Ziegler et al., 2010*; see Alpha and Beta Rhythms Tutorial text for details). (A) Here we are using the default HNN network configuration and not directly comparing the waveform to data, so begin with Step 3: activate the local network. In this example, a beta component emerged when the parameters of two ~ 10 Hz bursts to the local network through proximal and distal project pathways, as described in *Figure 7*, were adjusted so that on average they arrived to the network at the same time (see red boxes). This parameter set is provided in the 'AlphaAndBeta.params' file. (B) Step 4: running the simulation with the 'Run Simulation' button, shows that intermittent and transient alpha and beta rhythms emerge in the current dipole signal (middle dipole time trace; bottom time-frequency representation). Green and red histograms at the top display the defined distal and proximal burst drive patterns, respectively. Due to the stochastic nature of the bursts, on some cycles of the drive, the distal burst was simultaneous with the proximal burst and strong enough to push current flow down the dendrites to create a beta event (see red box). This model derived prediction reproduced several features of the data, including alpha and beta peaks in the corresponding PSD that were more closely matched to the recorded data (C). Model predictions were subsequently validated with invasive recordings in mice and monkeys (*Sherman et al., 2016*; see further discussion in text).

We begin by describing the process for simulating a pure alpha frequency rhythm only, and we then describe how a novel prediction for the origin of beta events emerged (*Sherman et al., 2016*). Motivated by a long history of research showing alpha rhythms in neocortex rely on ~10 Hz bursting in the thalamus, we tested the hypothesis that ~10 Hz bursts of drive through proximal and distal projection pathways (representing lemniscal and non-lemniscal thalamic drive) could reproduce an alpha rhythm in the local circuit. The burst statistics (number of spikes and inter-burst interval chosen from a Gaussian distribution), strength of the input (post-synaptic conductance), and delay between the proximal and distal input were manually adjusted until a pure alpha rhythm sharing feature of the data was found. We showed that when ~ 10 Hz bursts of proximal and distal drives are sub-threshold and arrive to the local network in anti-phase (~50 ms delay) a pure alpha rhythm emerges (*Jones et al., 2009*; *Ziegler et al., 2010*).

The parameters of this drive are distributed with HNN in the file 'Alpha.param', loaded through the Set Parameters From File button and viewed in the Set Parameters dialog box under Rhythmic Proximal and Rhythmic Distal inputs (*Figure 7A*). Note that the start time mean of the ~10 Hz Rhythmic Proximal and Rhythmic Distal Inputs are delayed by 50 ms. The HNN GUI in *Figure 7B* displays the simulated current dipole output from this drive (middle), the histogram of the proximal and distal driving spike trains (top), and the corresponding time-frequency domain response (bottom). This GUI window is automatically constructed when rhythmic inputs are given to the network, and HNN is designed to easily define rhythmic input to the network via the Set Parameters dialog box. A scaling factor was also applied to this signal (via Set Parameters, Run dialog box) and is shown as 300,000 on the y-axis of the main GUI window example in *Figure 7B*. The 300,000 scaling factor predicts that 60,000,000 PNs (300,000 × 200 PNs) contribute to the measured signal.

## Step 4

The alpha rhythm shown in *Figure 7B* is reproduced by clicking the 'Run Simulation' button at the top of the GUI, Additional network features, including power-spectral density plots, can also be visualized through the pull down menus (Step 5).

## Steps 6 and 7

Rhythmic input parameters can be adjusted to account for the user defined 'simulation experiment' and hypothesis testing goals.

The goal in our prior study was to reproduce the alpha/beta complex of the SI mu-rhythm. By hand tuning the parameters we were able to match the output of the model to several features of the recorded data, including symmetric amplitude modulation around zero and PSD plots as shown in *Figures 7* and *8* (see further feature matching in *Jones et al., 2009* and *Sherman et al., 2016*), we arrived at the hypothesis that brief bouts of beta activity ('beta events') non-time locked to alpha events could be generated by decreasing the mean delay between the proximal and distal drive to 0 ms and increasing the strength of the distal drive relative to the proximal drive. This parameter set is also distributed with HNN ('AlphaandBeta.param') and viewed in *Figure 8A*. With this mechanism, beta events emerged on cycles when the two stochastic drives hit the network simultaneously and when the distal drive was strong enough to break the upward flowing current and create a prominent ~50 ms downward deflection (see red box in *Figure 8B*). The stronger the distal drive the more prominent the beta activity (data not shown, see *Sherman et al., 2016*). This beta event hypothesis was purely model derived and was based on matching several features of the SI mu rhythm between the model output and data (detailed in *Jones et al., 2009 Sherman et al., 2016*).

Importantly, due to the non-time locked nature of this spontaneous rhythm, the waveform can not be directly compared by overlaying the waveform of the model and recorded oscillations as in the evoked response example (e.g. *Figure 4*). However, one can quantify features of the oscillation and compare to recorded data (*Jones et al., 2009*; *Sherman et al., 2016*). One such feature is the amplitude of the oscillation waveform, where a scaling factor can be applied to the model to predict how many cells are needed to produce a waveform amplitude on the same order as the recorded data, as described in Step three above. Additionally, the PSD from the model and data can be directly compared. This can be viewed in the HNN GUI though the 'View PSD' pull down menu (see *Figure 4*, Step 5), where this data ('SI_ongoing.txt' - provided with HNN) can be automatically compared to the model output in the PSD window (*Figure 8C*).

The model derived predictions on mechanisms underlying alpha and beta where motivated by literature and further refined by tuning the parameters to match the output of the model with various features of the recorded data. While the mechanisms of the alpha rhythm described above were motivated by literature showing cortical alpha rhythms arise in part from alpha frequency drive from the thalamus and supported by animal studies (*Hughes and Crunelli, 2005*; for example, see Figure 2 in *Bollimunta et al., 2011*), the beta event hypothesis was novel. The level of circuit detail in the model led to specific predictions on the laminar profile of synaptic activity occurring during beta events that could be directly tested with invasive recordings in animal models. One specific prediction was that the orientation of the current during the prominent ~50 ms deflection defining a beta event (red box, *Figure 8B*) was down the pyramidal neuron dendrites (e.g. into the cortex). This prediction, along with several others, were subsequently tested and validated with laminar recordings in both mice and monkeys, where it was also confirmed that features of beta events are conserved across species and recording modalities (*Sherman et al., 2016*; *Shin et al., 2017*).

## Gamma rhythms

Gamma rhythms can encompass a wide band of frequencies from 30 to 150 Hz. Here, we will focus on the generation of so-called 'low gamma' rhythms in the 30–80 Hz range. It has been well established through experiments and computational modeling that these rhythms can emerge in local spiking networks through excitatory and inhibitory cell interactions where synaptic time constants set the frequency of the oscillation, while broadband or 'high gamma' rhythms reflect spiking activity in the network that creates sharp waveform deflections (*Lee and Jones, 2013*). The period of the low gamma oscillation is determined by the time constant of decay of GABAA-mediated inhibitory currents (*Buzsáki and Wang, 2012*; *Cardin et al., 2009*; *Vierling-Claassen et al., 2010*), a

mechanism that has been referred to as pyramidal-interneuron gamma (PING). In normal regimes, the decay time constant of GABA$_A$-mediated synapses (~25 ms) bounds oscillations to the low gamma frequency band (~40 Hz). In general, PING rhythms are initiated by 'excitation' to the excitatory (PN) cells, and this initial excitation causes PN spiking that, in turn, synaptically activates a spiking population of inhibitory (I) cells. These (I) cells then inhibit the PN cells, preventing further PN activity until the PN cells can overcome the effects of the inhibition ~25 ms later. The pattern is repeated, creating a gamma frequency oscillation (~40 Hz).

We have applied HNN to determine if features in the current dipole signal could distinguish PING-mediated gamma from other possible mechanisms such as exogenous rhythmic drive or spiking activity that creates 'high gamma' oscillations (*Lee and Jones, 2013*). Here, we describe the process for generating gamma rhythms via the canonical PING mechanisms in HNN. First, to demonstrate the basic mechanisms of PING, we simulate a robust large amplitude and nearly continuous gamma rhythm (Steps 3–5, *Figure 9*). Second, we adjust the simulation parameters in order to directly compare to experimental data, where on single trials induced gamma rhythms are smaller in amplitude and less continuous emerging as transient bursts of activity (*Pantazis et al., 2018*) (Steps 6 and 7, *Figure 10*). Both parameter sets for creating the examples below are distributed with the software.

To demonstrate the robust PING-mechanisms and its expression at the level of a current dipole, we begin this tutorial with Step 3, 'activating' the network using a slightly altered local network configuration as described below.

## Step 3

'Activate' the local network by loading in the parameter set defining the local network and initial input parameters 'gamma_L5weak_L2weak.param'. In this example, the input was noisy excitatory synaptic drive to the pyramidal neurons. Additionally, all synaptic connections within the network are turned off (synaptic weight = 0) except for reciprocal connections between the excitatory (AMPA only) and inhibitory (GABA$_A$ only) cells within the same layer. This is not biologically realistic, but was done for illustration purposes and to prevent pyramidal-to-pyramidal interactions from disrupting the gamma rhythm. To view the local network connections, click the 'local network' button in the Set Parameters dialog box. *Figure 9B* shows the corresponding dialog box where the values of adjustable parameters are displayed. Notice that the L2/3 and L5 cells are not connected to each other, the inhibitory conductance weights within layers are stronger than the excitatory conductances, and there are also strong inhibitory-to-inhibitory (i.e., basket-to-basket) connections. This strong autonomous inhibition will cause synchrony among the basket cells, and hence strong inhibition onto the PNs.

To reproduce the ~40 Hz gamma oscillation described by the PING mechanism above, we drove the pyramidal neuron somas in L2/3 and L5 with noisy excitatory AMPA synaptic input, distributed in time as a Poisson process with a rate of 140 Hz. This noisy input can be viewed in the 'Set Parameters' menu by clicking on the 'Poisson Inputs' button (see *Figure 9A*). Setting the stop time of the Poisson drive to −1, under the Timing tab, keeps it active throughout the simulation duration.

## Step 4

The gamma rhythm shown in *Figure 9C* is reproduced by clicking the 'Run Simulation' button at the top of the main HNN GUI. The top panel shows a histogram of Poisson distributed times of input to the pyramidal neurons, the middle panel the net current dipole across the entire network, and the bottom the corresponding time-frequency spectrogram showing strong gamma band activity. Additional network features, including spiking activity in each cell in the population (*Figure 9D*), somatic voltages (*Figure 9E*), and PSD plots for each layer and the entire network (*Figure 9F*) can also be visualized through the 'View' pull down menu (Step 5). Notice the PING mechanisms described above in the spiking activity of the cells (*Figure 9D*), where in each layer the excitatory pyramidal neurons fire before the inhibitory basket cells. The line plots, which show spike counts over time, also demonstrate rhythmicity. The pyramidal neurons are firing periodically but with lower synchrony due to the Poisson drive (orange histogram at the top), which creates randomized spike times across the populations (once the inhibition sufficiently wears off). Notice also that the power in the gamma band is much smaller in Layers 2/3 than in Layer 5 (*Figure 9F*). This is reflective, in part, of the fact

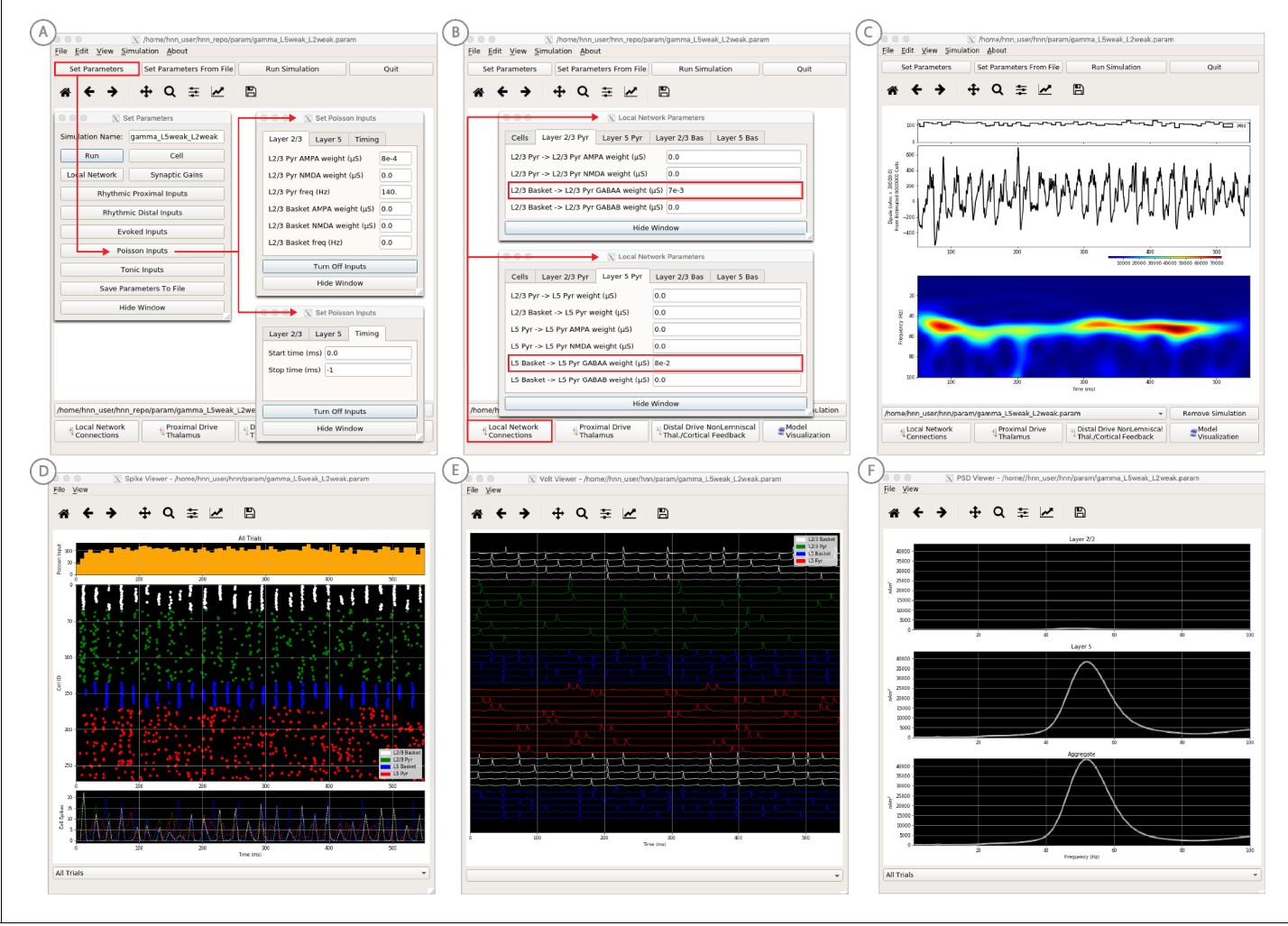

**Figure 9.** An example workflow for simulating canonical pyramidal-interneuron gamma (PING) rhythms (*Lee and Jones, 2013*; see Gamma Rhythms Tutorial text for details). (**A**) Here we are using the default HNN network configuration (with some parameter adjustments as shown in panel B) and we are not directly comparing the waveform to data, so begin with Step 3: activate the local network. Motivated by prior studies on PING mechanisms (see text), in this example PING rhythms were simulated by driving the pyramidal neuron somas with noisy excitatory synaptic input following a Poisson process. The parameters defining this noisy drive are viewed and adjusted through the Set Parameters button as shown, see text and Materials and methods for parameter details. This parameter set for this example is provided in the 'gamma_L5weak_L2weak.param' file. In this example, all synaptic connections within the network are turned off (synaptic weight = 0), except for reciprocal connections between the excitatory (AMPA only) and inhibitory (GABAA only) cells within the same layer. The local network connectivity can be viewed and adjusted through the Set Network Connection button or pull down menu, as shown in (**B**). (**C**) Step 4: running the simulation with the 'Run Simulation' button, shows that a ~ 50 Hz gamma rhythm is produced in the current dipole signal (middle dipole time trace; bottom time-frequency representation). The black histogram at the top displays the noisy excitatory drive to the network. (**D–F**) Step 5: additional network features, including cell spiking responses, somatic voltages, and layer specific power spectral density plots as shown can be visualized through the 'View' pull down menu.

that the length of the L2/3 PNs is smaller than the L5 PNs, and hence the L2/3 cells produce smaller current dipole moments that can be masked by activity in Layer 5. This example was constructed to describe features in the current dipole signal that could distinguish PING mediated gamma rhythms but has not been directly compared to recorded data (see *Lee and Jones, 2013* for further discussion).

## Steps 6 and 7

Next, we describe how parameter adjustments to the canonical PING mechanism above can account for induced gamma rhythms recorded with MEG in a prior published study (*Pantazis et al., 2018*).

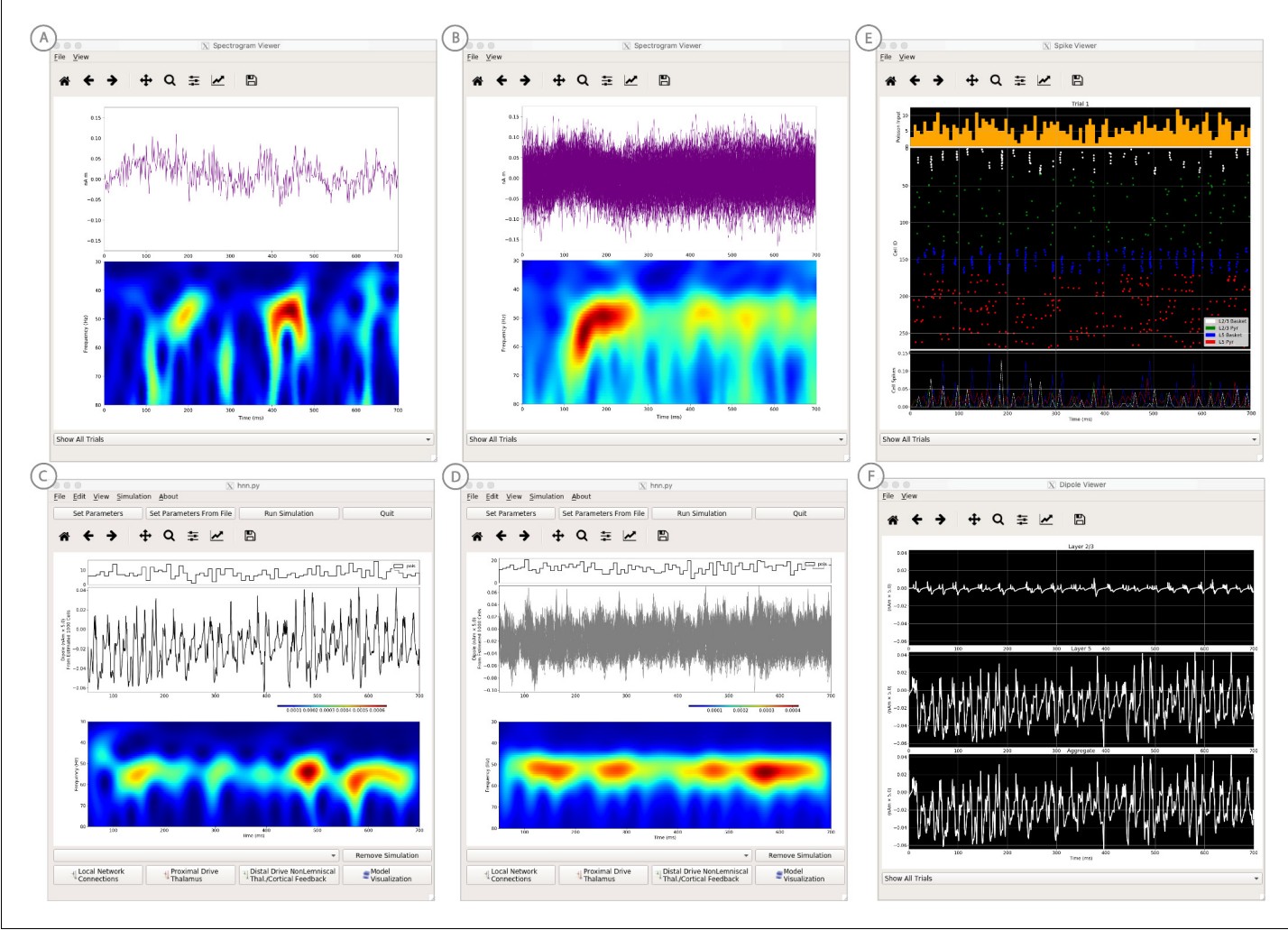

**Figure 10.** An example simulation describing how parameter adjustments to the canonical PING mechanism can account for source localized visually-induced gamma rhythms recorded with MEG. (**A**) An example single trial current dipole waveform generated from a stationary square-wave grating, presented from 0 to 700 ms. MEG data was source localized to the pericalcarine region of the visual cortex (top). The corresponding time-frequency representation (bottom) shows transient bursts of induced gamma activity (bottom). Data is as in *Pantazis et al. (2018)* and described further in text. (**B**) Analogous waveforms from 100 trials overlaid (top) and corresponding averaged time-frequency spectrograms (bottom). Averaging in the spectral domain creates the appearance of a more continuous oscillation. (**C**) Parameters used to simulate the canonical PING rhythm in *Figure 9* were adjusted (Steps 6 and 7) to test the hypothesis that a reduced rate of Poisson synaptic input to the pyramidal neurons could account for the bursty nature of the observed gamma data. The parameter set for this example is provided in the 'gamma_L5weak_L2weak_bursty.param' file. In this example, the Poisson input rates to the Layer 2/3 and Layer 5 cells were reduced to 4 Hz and 5 Hz, respectively, and the NMDA synaptic inputs were set to a small positive weight (dialog box not shown). Running the simulation will display a histogram of the Poisson drive (top), the current dipole waveform (middle), and the corresponding time-frequency spectrogram (bottom). With this parameter adjustment,~50 Hz bursty gamma activity is reproduced. A scaling factor of 5 was applied to the net current dipole to produce a signal amplitude comparable to the data (<0.1 nAm), suggesting that only ~1000 pyramidal neurons contributed the recorded signal. (**D**) Poisson input over 100 simulations (top), current dipole waveforms overlaid (middle), and the corresponding spectrogram average (bottom). Similar to the recorded data, averaging in the spectral domain creates the appearance of a more continuous oscillation. (**E–F**) Microcircuit details including cell spiking responses and layer specific current dipoles shows Layer 5 activity dominates the net current dipole signal, see text for further description.

The online version of this article includes the following source data for figure 10:

**Source data 1.** Visual cortex gamma activity from 1 trial.
**Source data 2.** Visual cortex gamma activity from 100 trials.

In this study, subjects were presented with the following visual stimuli: stationary square-wave Cartesian gratings with vertical orientation and black/white maximum contrast with a frequency of 3 cycles/degree. These stimuli create non-time locked induced gamma rhythms in the visual cortex; a reliable phenomena observed in many studies. The data represents time-domain activity from sources localized in the pericalcarine region of the visual cortex from 0 to 700 ms after the presentation of the stimulus (*Figure 10A,B*), and is located in HNN's data/gamma_tutorial folder.

To view the data in HNN, click on HNN's 'View' pull down menu, then click on 'View Spectrograms'. Then from the Spectrogram Viewer, click on the 'File' pull down menu, click 'Load Data', and select data/gamma_tutorial/100_trials.txt. *Figure 10A* shows a single trial of the source-localized current dipole signal, with corresponding wavelet-transform spectrogram below. Here, visual gratings were provided from time = 0–700 ms. On single non-averaged trials, the visual stimulus creates non-time-locked induced gamma rhythms at ~50 Hz that are not continuous but transient in time and 'burst-like', with each burst lasting ~50–100 ms. HNN's Spectrogram Viewer automatically averages the wavelet spectrograms across trials when the data is loaded (*Figure 10B*). To view a single trial as in *Figure 10A*, use the drop-down menu at the bottom of the Spectrogram Viewer to select a single trial. The averaged spectrogram from 100 trials makes the oscillation appear nearly continuous in the 40–50 Hz gamma band (*Figure 10B*).

We hypothesized that in order to change the nearly continuous gamma oscillation from HNN's canonical PING model (*Figure 9*) into a bursty lower amplitude gamma rhythm on single trials, the noisy excitatory AMPA synaptic input to the pyramidal neurons that creates the PING-rhythm, as described in Step 3, would need to be reduced. We tested this hypothesis by reducing the rate of the Poisson synaptic inputs to L2/3 and L5 pyramidal neurons, here from 140 Hz to 4 Hz and 5 Hz, respectively. To see the parameter values used, load the 'gamma_L5weak_L2weak_bursty.param' file, then click on 'Set Parameters' and 'Poisson Inputs' (parameter values not shown in *Figure 10*). Note, that in addition to the change and rate of drive, the NMDA synaptic inputs now have a small positive weight (previously zero), which was required to compensate for the reduced AMPA activation of the pyramidal neurons.

Click on 'Run Simulation' to produce the results shown in *Figure 10C*. The top panel of *Figure 10C* shows the Poisson inputs, which have a noticeably lower rate than the default PING simulation. The current dipole time-course is shown in the middle panel. The corresponding wavelet spectrogram in the bottom panel shows intermittent gamma bursts recurring with high power, similar to the features seen in the experiment. Note, in this data set, the amplitude of the current dipole signal is small,<0.1 nAm. As such, a small dipole scaling factor of 5 was applied to the output of HNN to compare to the amplitude of the experimental data. This predicts that a highly localized population of ~1000 pyramidal neurons contribute to the recorded gamma signal.

To see the effects of averaging across trials in HNN, click on 'Set Parameters' and 'Run', enter 100 trials, and then click on 'Run Simulation'. Output as shown in *Figure 10D* will be produced. The individual current dipole waveforms do not have consistent phase across trials (*Figure 10D* middle). However, averaging the spectrograms across trials produces a more continuous band of gamma oscillation throughout the simulation duration, similar to the effects observed in the experiment.

We can now use HNN to take a closer look at the underlying neuronal spiking activity contributing to the observed dipole signal, as shown in *Figure 10E*. The firing rates of L2/3 pyramidal neurons (green) and L5 pyramidal neurons (red) are significantly lower and more sparse than in the continuous PING simulation shown in *Figure 9*. This lower pyramidal neuron activation leads to fewer, or no basket interneurons firing on any given gamma cycle (*Figure 10E* white, blue points have fewer participating interneurons, and do not always occur). This lower interneuron activity produces lower-amplitude bouts of feedback inhibition, which are sometimes nearly absent, and thus creates transient bursty gamma activity. Note that the firing rates of L2/3 pyramids are lower than that of L5 pyramidals, consistent with experimental data (*Naka et al., 2019*; *Schiemann et al., 2015*) and previous data-driven modeling (*Dura-Bernal et al., 2019*; *Neymotin et al., 2016*). Layer specific current dipole responses in *Figure 10F* confirm that due to their dendritic length L5 pyramidals are once again the major contributors to the aggregate dipole signal. This example is presented as a proof of principle of the method to begin to study the cell and circuit origin of source localized gamma band data with HNN. The circuit level hypotheses have not yet been published elsewhere or investigated in further detail.

## ERP model optimization

To ease the process of narrowing in on parameter values representing a user's hypothesized model, we have added a model optimization tool in HNN. Currently, this tool automatically estimates parameter values that minimize the error between model output and features of ERP waveforms from experiments. Parameter estimation is a computationally demanding task for any large-scale model. To reduce this complexity, we have leveraged insight of key parameters essential to ERP generation, along with a parameter sensitivity analysis, to create an optimization procedure that reduces the computational demand to a level that can be satisfied by a common multi-core laptop.

Two primary insights guided development of the optimization tool. First, exogenous proximal and distal driving inputs are the essential parameters to first tune to get an initial accurate representation of an ERP waveform. Thus, the model optimization is currently designed to estimate the parameters of these driving inputs defined by their synaptic connection strengths, and the Gaussian distribution of their timing (see dialog box in *Figure 11B*). In optimizing the parameters of the evoked response simulations to reproduce ERP data distributed with HNN (e.g. see ERP tutorial), we performed sensitivity analyses that estimated the relative contributions of each parameter to model uncertainty, where a low contribution indicated that a parameter could be fixed in the model and excluded from the estimation process to decrease compute time (see Supplementary Materials).

Second, an intuitive insight that was confirmed by parameter sensitivity analysis is that the influence of each exogenous input on the simulated dipole varies over time, with the highest influence during and just after the time of the input (see Supplementary Material). We used this knowledge to create a stepwise optimization process, only estimating parameter values for one input at a time, where the objective of each optimization is to minimize a weighted root mean squared error (RMSE) measure between simulated and experimental data only during the relevant time window (see Materials and methods). This stepwise estimation reduces the complexity of the optimization problem and saves time. Each step in the process searches for parameter estimates using the COBYLA optimization algorithm (*Powell, 1994*) (see Materials and methods for detailed explanation of the stepwise optimization procedure).

## Example model optimization for the suprathreshold sensory evoked response data set

In this example, we describe an application of the model optimization tool for estimating parameters to simulate data representing the SI evoked response to a brief suprathreshold level tactile stimulation – which is 100% detected (*Figure 11A*). This evoked response is similar to that shown in *Figure 4*, where the signal was elicited from a perceptual threshold level stimulation - at 50% detection. We start from the parameter file fitted to the 50% detection scenario, and use HNN's model optimization feature to find parameter estimates that provide a better fit the suprathreshold-level experimental data. The data from this study is also included in the HNN distribution ('SI_SupraT.txt').

### Steps 1–4

Similar to steps 1–4 above, first load the supra-threshold experimental data file 'S1_SupraT.txt' via the 'Load data file' menu option and the example starting parameters to activate the network provided in the parameter file 'ERPYes100Trials.param' via the 'Load parameter file' menu option. Note that in this example, the network is also 'activated' by a sequence of three exogenous inputs defined in the parameter file. The parameters for these inputs serve as a baseline for model optimization. The supplied parameter file (used above) runs 100 trials by default for each simulation. For model optimization, this can be reduced to three trials. Click on the 'Set Parameters' button, then the 'Run' button, and replace 100 trials with 3. In the previous Set Parameters dialog box change the simulation name to 'ERPYes3Trials' to reflect this change (*Figure 4C*). By clicking the 'Run Simulation' button the evoked response using this initial parameter set as in *Figure 11A* will be displayed. As described above, in practice with user defined data, users should apply their own hypotheses related to the number, timing and synaptic input strengths of the exogenous inputs that activate the network to obtain an initial representation of the recorded waveform before beginning the parameter estimation process.

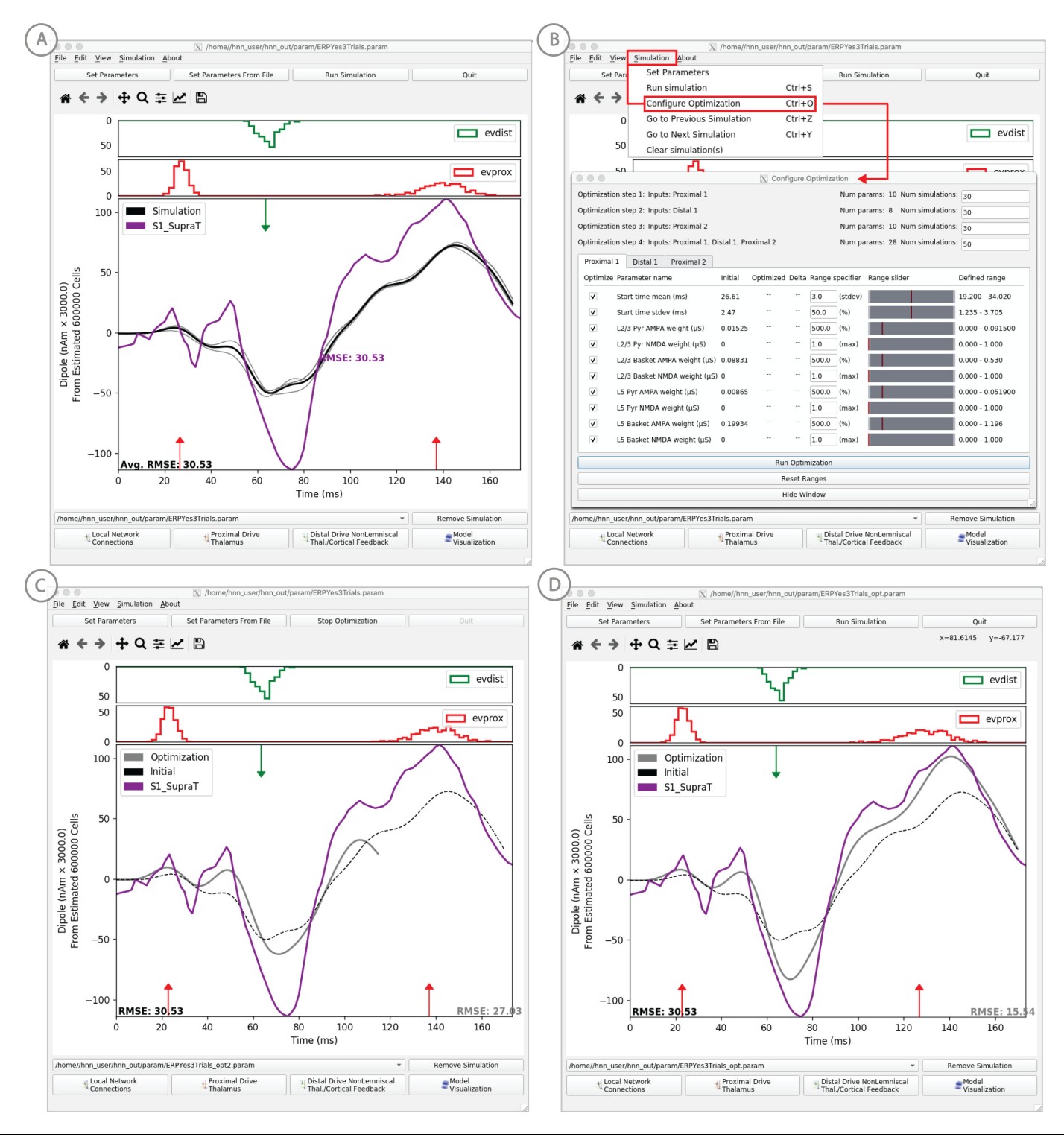

**Figure 11.** Example of the ERP parameter optimization procedure for a suprathreshold tactile evoked response. (**A**) Source localized SI data from a suprathreshold tactile evoked response (100% detection; purple) is shown overlaid with the corresponding HNN evoked response (black) using the threshold level evoked response parameter set detailed in *Figure 4*, as in initial parameter set. The RMSE between the data and the model is initially high at 30.53. (**B**) To improve the fit to the data, a procedure for sequentially optimizing the strengths of the proximal and distal drive inputs generating the evoked response can be run. The 'Configure Optimization' option is available under the 'Simulation' pull down menu. A dialog box allows users to choose and set a range over free optimization parameters, see text for details. (**C**) The GUI displays an intermediate fit after the first optimization step, specific to the first proximal drive. (**D**) The final fit is displayed once the optimization is complete. Here, the simulation from the optimized parameter

*Figure 11 continued on next page*

*Figure 11 continued*

set for the suprathreshold evoked response is shown in gray with an improved RMSE of 15.54 compared to 30.53 for the initial model. See *Figure 11—figure supplements 1* and *2* for a further description of the optimization routine and parameter sensitivity analysis.

The online version of this article includes the following source data and figure supplement(s) for figure 11:

**Source data 1.** Average SI ERP from suprathreshood stimulus.
**Figure supplement 1.** Weighted scoring of the stepwise optimization procedure for suprathreshold level evoked response example, see text for details.
**Figure supplement 2.** Sensitivity analysis results of the suprathreshold level evoked response example showing the relative contribution of each input's parameters on variance.

## Step 5

Before running the optimization, rename the simulation to 'ERPYes3Trials_opt' in the Set Parameters dialog box as described above, so that the parameter results of the optimization will be saved in a new file.

## Step 6

In the Simulation pull down menu, choose the 'Configure Optimization' option. This option is only selectable once data and parameter files have been loaded. A new dialog box pre-populated with values from the parameter file will appear, as shown in *Figure 11B*. All parameters describing the timing and strength of defined exogenous inputs will be available for optimization. Users can generate their own evoked response parameter files with as many exogenous inputs as desired and they will be automatically populated into the 'Configure Optimization' dialog box.

Select which parameters to treat as free variables for optimization; parameters that will be fixed in the optimization process are grayed out. By default, all parameters are selected, but it may be desirable to limit the number of free parameters to only the most influential set based on a parameter sensitivity analysis. Fixing non-influential parameters will decrease the complexity of an optimization step, and increase the likelihood of the optimization algorithm converging on parameter estimates after a relatively low number of simulations. Results of a sensitivity analysis using Uncertainpy (*Tennøe et al., 2018*) on this example data are provided in *Supplementary file 1* and may help guide model optimization for similar data. Sensitivity analysis is not yet included in HNN (see Future Directions).

The number of simulations per optimization step is configurable in the top section of the 'Configure Optimization' dialog box (*Figure 11B*). The default values shown in *Figure 11B* were based on results from our studies where the fit obtained was significantly improved from a single optimization. This value can be decreased as the number of free parameters is reduced.

The parameter ranges defining the bound constraints given to the optimization algorithm are shown in the 'Defined range' column of the dialog box in *Figure 11B*. The displayed range is calculated as plus or minus a specified number of standard deviations for input start time or plus or minus a percentage of the initial value for all other parameters. The user may customize the range by inputting their own 'Range specifier' or using the interactive slider bar to define new minimum or maximum values. If a parameter has an initial value of 0, its range is defined by a user-specified maximum value rather than percentage. The 'Reset Ranges' button will update ranges using the 'Range specifier' and discard custom values set by the slider. Parameters can be fixed during the optimization process by unchecking the 'Optimize' checkbox.

## Step 7

Click the 'Run Optimization' button to start the stepwise optimization process. After each input has been optimized in sequence followed by a final optimization step that adjusts all input parameters, the final optimized fit will be shown in gray in the main HNN window along with the lowest obtained RMSE (*Figure 11C/D*).

## Step 8 (optional)

To perform a second optimization using the results of the first procedure as a starting point, select the optimized simulation parameter set drop-down menu. This will update the values in the

Configure Optimization dialog box and pressing Run Optimization will start a new optimization process. For this example, the RMSE improved from 15.54 (*Figure 11D*) after the first optimization to 10.02 after a second round (data not shown).

## Discussion

The Human Neocortical Neurosolver (HNN, https://hnn.brown.edu) is a neural modeling software tool developed to help researchers and clinicians interpret the neural origin of their human EEG or MEG data. HNN's interactive GUI is designed for users with no formal computational neural modeling or software development experience to be able to develop and test hypotheses on the cellular- and circuit-level generators of their human data. Based on prior applications of HNN's underlying template neural model on these signals (*Jones et al., 2009*; *Jones et al., 2007*; *Khan et al., 2015*; *Lee and Jones, 2013*; *Sherman et al., 2016*; *Sliva et al., 2018*; *Ziegler et al., 2010*), the tutorials and the example workflow focus on studying the neural origin of ERPs and low-frequency oscillations from a single brain region. The template network model contains features of a canonical neocortical circuit, with layer-specific thalamocortical and cortico-cortical drive, where the net primary current dipoles are simulated from the intracellular current across the network of pyramidal neurons. HNN enables visualization and direct comparison of the primary current dipole produced by the network to source-localized data in units of Am, under various parameter manipulations. This comparison, along with simultaneous visualization of microcircuit activity, including cell spiking and somatic voltage responses, guides interpretation of the cellular- and circuit-level origin of EEG/MEG data.

HNN was created based on the biophysical origin of EEG/MEG primary currents to be a hypothesis development and testing tool, where specific predictions on the microcircuit-level underpinnings of recorded data can be produced. The circuit-level predictions can guide further validation with invasive recordings or with other imaging modalities (e.g., spectroscopy or tractography, see *Khan et al., 2015*). As one specific example, HNN led to a novel prediction on the origin of transient neocortical beta oscillations, and the prediction was later tested and supported by laminar recordings in mice and monkeys (*Sherman et al., 2016*). In turn, established cellular- or circuit-level details known to contribute to healthy brain dynamics and/or disease states can be adapted into HNN to predict corresponding signatures in macroscale signals.

HNN is particularly timely given the rapidly expanding wealth of genetic insights and phenotype data in animal model systems. As disease-specific genetic mutations and corresponding cellular/circuit outcomes in mouse models are identified, they can be implemented in HNN, and their impact on EEG/MEG measured brain dynamics, ranging from ongoing state properties (e.g., alpha oscillations) to sensory-evoked responses, can be simulated. The outputs from HNN would then provide specific and principled predictions to be compared against real EEG/MEG data obtained in the relevant population, leading to valid bi-directional inference. Overall, the scalability of HNN provides an unprecedented framework for translational neuroscience research.

### Comparison to other neocortical models and EEG/MEG modeling software

Many models of neocortical circuitry, with varying levels of complexity, have been developed to simulate LFP, EEG/MEG and/or ECoG (e.g., *Barrès et al., 2013*; *Kiebel et al., 2008*; *Reimann et al., 2013*; *Sanz Leon et al., 2013*). Several modeling tools and associated documentation are also available to build user defined neocortical models for general use that are not domain specific, such as NEURON (https://neuron.yale.edu/neuron/), NetPyNE (*Dura-Bernal et al., 2019*), the Brain Modeling Toolkit/Bionet (*Gratiy et al., 2018*), and the Brain Simulation Platform from the European Union Human Brain Project. Among the current modeling software designed specifically for study of EEG/MEG signals (e.g., The Virtual Brain [TVB] https://thevirtualbrain.org, Dynamic Causal Modeling [DCM] of E/MEG within the Statistical Parametric Mapping [SPM] software https://www.fil.ion.ucl.ac.uk/spm/, and LFPy https://lfpy.readthedocs.io/en/latest/; *Hagen et al., 2018*; *Kiebel et al., 2008*; *Sanz Leon et al., 2013*), HNN's model, goals, and capabilities are unique.

The goal of HNN is to provide a user-friendly graphical interface to a validated biophysically detailed model of neocortical circuitry, and to teach the community, regardless of neural modeling or coding experience, how to interact with the model to study the neural origin of commonly measured macroscale EEG/MEG signals. This includes studying ERPs, and low frequency alpha, beta and

gamma rhythms. HNN's construction and tutorials are based on knowledge and workflows developed in prior published studies. As with other open-source software, continued application of HNN to new use cases means that software users can add to and improve upon the examples distributed with HNN. The level of biophysical detail included in HNN's model and the calculation of the primary electrical currents from the intracellular dendritic current flow in multi-compartment pyramidal neurons enables one-to-one comparison between model output and source localized data in units of Am. HNN was specifically designed for interpreting microscale cellular- and circuit-level activity from single regions of interest. The cell and network level details provided can guide targeted testing and make connections to studies in animal models. Below we describe ways in which HNN's goals and construction are distinct from other current domain specific EEG/MEG modeling software, namely LFPy, TVB, DCM.

LFPy is a Python package that provides a set of Python libraries and associated documentation on how to apply these scripts to simulate multi-scale signals, including current dipole, LFP, ECoG, M/EEG sensor signals, in user defined multi-compartment neuron models and networks built in NEURON or NeuroML (*Hagen et al., 2018*). LFPy does not contain a GUI and is designed for users who have experience in neural modeling and Python. Users define their own workflows to simulate signals of interest that can be compared to data. The LFPy Python classes are likely to provide a useful framework for expanding the utility of HNN to include multi-area simulations, and simulations of LFP and EEG/MEG sensor level signals, as described in Limitations and future directions below.

DCM applied to EEG/MEG data is also a non-GUI based scripting tool, using Matlab. Users assume an active set of distributed sites, that is nodes, in the brain that contribute to a recorded signal. The neural activity of a node is simulated using 'neural mass' representations in which the activity (e.g. firing rate) of a population of neurons is simulated with a reduced number of variables (*Kiebel et al., 2008*). The recorded data is fit to the assumed nodes and directionality of interactions between nodes statistically inferred.

TVB is designed to simulate large-scale network interactions also using reduced neural mass representations. Active nodes across the whole brain are assumed to contribute to the recorded signal and connectivity between nodes is informed by individualized tractography data (*Sanz Leon et al., 2013*). Multi-scale EEG/MEG and/or fMRI data can be fit to the model. One advantage of this approach is that propagation of activity across the brain can be studied (e.g. spread of seizure), unlike HNN which is currently restricted to interpreting detailed activity in a single region of interest.

Indeed, many prior models of EEG/MEG rely on reduced representations of neural activity, including neural mass and/or mean field approximations (*Breakspear et al., 2004*; *Jansen and Rit, 1995*; *Jirsa and Haken, 1996*; *Kiebel et al., 2008*; *Sanz Leon et al., 2013*; *Woolrich and Stephan, 2013*). Such simplifications may be necessary to ensure mathematical or computational tractability of models that address interactions between multiple areas or whole brain activity (*Breakspear, 2017*). However, that tractability comes at the cost of suppressing or eliminating the ability to evaluate cellular-level details of individual spiking units and dendritic currents, or to perform one-to-one comparisons between model and data; explicit goals of HNN.

## Limitations and future directions

One of the greatest challenges in computational neural modeling is deciding the appropriate scale of model to use to answer the question at hand. There is always a tradeoff between model complexity and computational efficiency, ease of use, and interpretability. As discussed above, this tradeoff underlies different scales of modeling in various EEG/MEG modeling software. HNN's model was chosen to be minimally sufficient to accurately account for the biophysical origin of the primary currents that underlie EEG/MEG signals in a single brain area; namely, the net intracellular current flow in the apical dendrites of pyramidal neurons that span across the cortical layers and receive layer-specific synaptic input from other brain areas. HNN's model was also constructed to maintain known canonical features of neocortical circuitry including, excitatory/inhibitory ratios, layer specific synaptic interactions, and cell spiking behaviors (see Parameter Tuning above, and Materials and methods). While HNN's model is a reduction of the full complexity of neocortical circuits, it has been successful in interpreting the origin of extracranially measured macro-scale EEG/MEG signals that likely rely on canonical macroscale features of neocortical circuitry and not on finer details of the underlying structure. A future direction discussed below is to expand HNN to simulate extracellular local field potential signals (LFPs), and sensor level signals, whose accuracy may require additional model detail and

whose implementation can be aided by other existing tools such as LFPy and NetPyNe (discussed further below). Any conclusions made with HNN are based on the underlying model assumptions that are important for users to understand. These assumptions are outlined in detail in the Materials and methods section, in our prior publications, and on our website.

Parameter optimization is a computationally challenging problem in any large-scale model. The process for parameter tuning to study ERPs and oscillations in HNN's underlying model is detailed above. Based on our prior studies and sensitivity analyses (see Supplementary Materials), we have identified that the timing and strength of the layer specific exogenous drive to the local network is critical in defining the timing and peaks of sensory evoked responses. As such, HNN currently includes a tool to optimize these parameters based on reducing the error between simulated evoked response waveforms and recorded data. Due to the non-stationary nature of spontaneous brain rhythms (e.g. *Figure 6* and *Figure 10*) error reduction based on matching waveform features is not as straightforward, and other signal features may be necessary to consider for optimization (e.g. PSD peak amplitudes, see *Figure 8* and *Jones et al., 2009*). Future expansions of HNN will include the ability to optimize over other user defined parameters, and to minimize errors between model output and various features of recorded data, with an estimate of the sensitivity of various parameters to these features. Given enough compute power, large parameter sweeps could be implemented in HNN to generate families of models for template matching to given waveforms via machine learning algorithms. This would serve as an alternative means for circuit interpretation without interactive hypothesis development and testing. At present, HNN can be run on high performance computers through the Neuroscience Gateway Portal (www.nsgportal.org) and Amazon Web Services (https://aws.amazon.com), see also Dissemination in Materials and methods.

Currently, all conclusions made in HNN are derived from the template neocortical column model provided. Another important step in expanding HNN's utility will be to enable users to define their own cells and circuits to use within the HNN framework. While the HNN code is open source and adaptable for advanced users, it is difficult for those without expertise in computational neural modeling in Neuron/Python to expand. Therefore, work is in progress to convert HNN's underlying neural model to the NetPyNe simulation language (www.netpyne.org) (*Dura-Bernal et al., 2019*). NetPyNe is a neural modeling platform enabling flexible cell and network development. This conversion will facilitate the ability to expand HNN to the study of activity from and between multiple cortical areas and the thalamus. NetPyNe is designed with both a GUI and command line (CLI) interface facilitating the construction of code that is readily accessible and human-readable. In expanding HNN to the NetPyNe language, HNN will also embrace the dual GUI and CLI capabilities, enabling the specification of architectures and parameters to be scriptable so that simulations and analyses can scale-up beyond manual operations.

HNN is designed to simulate source-localized current dipole signals produced by neurons. Source localization is currently viewed as an independent process. The output from any source localization algorithm can be compared to HNN's simulated output. In future expansions of HNN, we plan to integrate HNN's 'bottom up' simulations, with 'top down' source localization estimates using minimum-norm-estimate (MNE) software (www.martinos.org/mne) (*Gramfort et al., 2013*; *Gramfort et al., 2014*), providing an all-in-one software tool for source localization and circuit-based interpretation. In doing so, parameter estimation in each software package may benefit from direct knowledge and constraints from the other. Additionally, HNN's utility will be expanded to include estimation of forward fields through the brain to simulate and visualize LFPs, current-source density, and sensor-level EEG/MEG signals, facilitating comparison to these recording modalities.

We have shown that HNN can be a useful tool to interpret the impact of noninvasive brain stimulation (NIBS) on EEG-measured circuit dynamics (*Figure 5*, *Sliva et al., 2018*). HNN was used to test specific hypotheses on tACS-induced modulation of synaptic dynamics by accounting for EEG signal differences in pre-tACS compared to post-tACS periods. A useful expansion of HNN will be to include simulations of the fields induced in the brain by NIBS (e.g., with finite-element-estimates *Windhoff et al., 2013*) and to directly couple these fields to the modeled neurons. This integration would facilitate studying the effects of NIBS on real-time EEG signals and could lead to improved NIBS paradigms.

In total, HNN's present distribution and planned expansions are aimed at providing a one-of-a-kind, user-friendly software tool for translational neuroscience research that is accessible to a wide scientific and clinical community.

# Materials and methods

**Key resources table**

| Reagent type (species) or resource | Designation | Source or reference | Identifiers | Additional information |
|---|---|---|---|---|
| Software, algorithm | HNN | HNN | RRID:SCR_017437 | https://hnn.brown.edu |
| Software, algorithm | NEURON | NEURON | RRID:SCR_005393 | https://neuron.yale.edu |

*Dissemination* HNN is distributed online at https://hnn.brown.edu. The menu bar at the top of the HNN homepage links to installation instructions, documentation, tutorials, troubleshooting information, and a user forum. HNN can be installed locally on Linux, Windows, and mac OS operating systems, and it can be run as well online through Amazon Web Services (AWS) or the Neuroscience Gateway Portal (NGP). Since HNN is an open-source project, the code for our software, as well as the local installation instructions, are hosted on GitHub (see https://github.com/jonescompneurolab/hnn).

## Template model construction
### Overview
The template neocortical model provided with HNN is based on prior publications using the model without a graphical user interface, as described in *Jones et al. (2009)*, and available on ModelDB (https://senselab.med.yale.edu/modeldb). All current parameter files included in the software, and described in the tutorials above, are based on this model, except for the gamma tutorial whose parameter file has local network modifications as described above.

HNN's underlying neocortical model is simulated using the NEURON simulation environment with the Python interpreter (see Key Resources Table). HNN's model is simulated across multiple cores in parallel using the message-passing interface (MPI). HNN's Run Parameters dialog box can be accessed through the GUI and provides access to commonly used simulation parameters, including integration time-step (dt), simulation duration (milliseconds), number of trials, neuronal firing threshold (mV), and number of cores over which to parallelize the model.

The model represents a canonical neocortical circuit. It contains multi-compartment pyramidal neurons (PN) in supragranular and infragranular layers (layers 2/3 and 5, respectively), whose apical dendrites are spatially aligned and span the cortical layers. In both layers, the PNs have two basal, one oblique, and one apical dendrite branch, and the layer 2/3 PNs have shorter apical dendrites than layer 5 PNs.

The PNs are synaptically coupled to each other and to a subset of inhibitory neurons in each layer, and are included in the model in a 3/1 PN-to-interneuron ratio, with a scalable number of PNs. The inhibitory neurons are simulated with single compartments representing fast spiking basket cells, and are shown in yellow in (*Figure 2*). Note that the granular layer is not explicitly included in the template circuit. This design choice was based on the fact that macroscale current dipoles are dominated by PN activity in supragranular, and infragranular layers, due to their alignment (see Calculation of Primary Electrical Current). Thalamic input to granular layers is presumed to propagate directly to basal and oblique dendrites of PN in layer 2/3 and 5.

## Detailed neuronal morphology and physiology
### Morphology
The morphology of the PN in each layer (see *Figure 3*, and *Table 1*) were adapted from the morphology reduction procedure in *Bush and Sejnowski (1993)*. In *Bush and Sejnowski (1993)*, digitized HRP-filled pyramidal neurons from layers 2 and 5 of cat visual cortex consisting of 400 compartments were reduced to 8 and 9 compartments, respectively. This procedure is important for HNN's purposes because the axial conductance is the basis of the primary current dipole calculation (see below).

The reduction procedure was based on conserving the axial conductance between compartments in the cells rather than the surface area of the dendritic tree. This reduction retained the general morphology of the neurons and allowed for accurate position of synaptic inputs and ionic conductances on individual cells, and construction of spatially accurate network models. The reduced model

**Table 1.** Length ($\mu m$), diameter ($\mu m$) of compartments in each modeled neuron type.

Adend (Bdend) represent apical (basal) dendrite. Note the following connectivity for compartments of PNs. The non-oblique Adends of PNs are connected vertically along the Z axis (cortical layer axis from supra- to infragranular layers) from soma → Adend trunk → Adend1 → Adend2 → Adend tuft. The smaller L2/3 PNs do not have an Adend2 compartment. PN Adend oblique are connected to the soma and perpendicular to the Z axis. Bdend1 connects to the soma along the Z axis, and Bdend2 and Bdend3 branch from Bdend1 at a 45 degree angle from the Z axis. L2/3 and L5 basket interneurons have a single somatic compartment. N/A indicates non-applicable, since that specific compartment not present in the neuron type. Geometry illustrated in Figure 3 above.

| Type | Soma | Adend trunk | Adend1 | Adend2 | Adend tuft | Adend oblique | Bdend1 | Bdend2 | Bdend3 |
|------|------|-------------|--------|--------|------------|---------------|--------|--------|--------|
| L2/3 PN | 22.1, 23.4 | 59.5, 4.3 | 306.0, 4.1 | N/A | 238.0, 3.4 | 340.0, 3.9 | 85.0, 4.3 | 255.0, 2.7 | 255.0, 2.7 |
| L2/3 basket | 39.0, 20.0 | N/A | N/A | N/A | N/A | N/A | N/A | N/A | N/A |
| L5 PN | 39.0, 28.9 | 102.0, 10.2 | 680.0, 7.5 | 680.0, 4.9 | 425.0, 3.4 | 255.0, 5.1 | 85.0, 6.8 | 255.0, 8.5 | 255.0, 8.5 |
| L5 basket | 39.0, 20.0 | N/A | N/A | N/A | N/A | N/A | N/A | N/A | N/A |

contained active conductances in the somatic compartments and retaining faithful electrical responses to injected current. In HNN, a scaling factor of 1.3 was applied to the length and diameters of the dendritic compartments to account for increases in dendritic length and volume in human somato-sensory neurons, as predicted by larger cortical thickness (*Fischl and Dale, 2000*; *Geyer et al., 1997*) and an increase in the number of dendritic spines and arborization (*Elston et al., 2001*). Additional active ionic conductances were added to the dendritic compartments, as detailed below, and the neurons were embedded in a cortical column network with single compartment inhibitory neurons. The radial geometry of the inhibitory neuron dendrites does not contribute to the primary current dipole calculation. The morphology of each cell was as follows:

Layer 2/3
- PN: 8 compartments including four apical dendrites, three basal dendrites, one soma
- Inhibitory basket neurons: single compartment (soma)

Layer 5

- PN: 9 compartments including five apical dendrites, three basal dendrites, one soma.
- As shown below, L5 PNs have longer dendrites than L2/3 PNs. L5 PN somas are based in L5 with long apical dendrites reaching into L2/3.
- Inhibitory Basket neurons: single compartment (soma).
- L2/3 and L5 basket interneurons are identical but their synaptic parameters and local circuit connectivity differs.

## Physiology

Membrane voltages in each simulated compartment were calculated using the standard Hodgkin-Huxley parallel conductance equations, and current flow between compartments follows from cable theory as accounted for in NEURON.

Extending the prior work of *Bush and Sejnowski (1993)*, active ionic currents were included in both the somatic and dendritic compartment of the cells of the pyramidal neurons, and in the single compartment of the inhibitory neurons. For the pyramidal neurons, the membrane resistance was increased and membrane capacitance was decreased from the Bush and Sejnowski's values by the same 1.3 scaling factor as the compartment sizes described above ($R_m$ 23,474 $cm^2$ for L5 and L2/3; $C_m$ 0.85 and $C_m$ 0.6195 F/cm2 for L5 and L2/3, respectively) to maintain the input resistances in the cells of 45 M for the L5 and 110 M for L2/3 (*Douglas et al., 1991*). The axial resistance for each cell was $R_a$ 200 cm (*Segev et al., 1992*). The parameters regulating the active currents were tuned to replicate known in vitro firing patterns in response to somatic current injection.

The kinetic equations and NEURON code used for each of these currents were as used by *Mainen and Sejnowski (1996)* and downloaded from http://senselab.med.yale.edu/senselab/mod-eldb/. The maximal conductances of each current were constant throughout the soma and dendrite (*Bekkers, 2000*; *Korngreen and Sakmann, 2000*; *Migliore and Shepherd, 2002*; *Stuart and Sakmann, 1994*) and were chosen to produce adapting spikes in the L2/3 PNs and bursting in the L5 PNs to current injected in the soma (1 nA for 100 ms) representative of neurons classified as regular

spiking and intrinsically bursting, respectively (*Moore and Nelson, 1998*; *Silva et al., 1991*; *Zhu and Connors, 1999*).

The inhibitory neurons were tuned to represent basket cells and produced regular fast spiking dynamics to injected current, as in other cortical network models (*Garabedian et al., 2003*; *Jones et al., 2000*; *Pinto et al., 2003*).

The following table displays the ion channels and mechanisms in each cell type in the model (X) indicates the presence of the channel/mechanism in the cell type, see online code for full equations.

| Cell type (rows) Channel/mechanism Type (columns) | Na (fast) | K (fast) | Km | KCa | Ca (L-type) | Ca (T-type) | Ca decay | HCN | Leak | Dipole |
|---|---|---|---|---|---|---|---|---|---|---|
| Basket | X | X | | | | | | | X | |
| L2/3 Pyramidal | X | X | X | | | | | | X | X |
| L5 Pyramidal | X | X | X | X | X | X | X | X | X | X |

In the table above, Na (fast)/K (fast) are the fast sodium and potassium channels responsible for generating action potentials. Km is the muscarine sensitive potassium channel, with a relatively slow time-constant and KCa is the calcium-dependent potassium channel, which contributes to hyperpolarization after calcium influx into the cell. The L- and T-type calcium (Ca) channels represent the high-threshold and low-threshold activated calcium channels which together with the hyperpolarization-activated cyclic nucleotide gated channel (HCN) contribute to bursting. Ca decay represents the calcium extrusion pump, which causes intracellular calcium to decay towards a baseline level. Leak represents the passive channel, with constant conductance. Dipole represents the mechanism that takes into account the primary axial current flow within pyramidal neuron dendrites, responsible for the generation of simulated signals comparable to MEG/EEG recordings. For more details see *Jones et al. (2009)*.

## Local network connections

HNN's default template neocortical model includes neurons arranged in three dimensions. The *XY* plane is used to array cells on a regular grid while the Z-axis specifies cortical layer. HNN's default model contains a regular $10 \times 10$ grid (arbitrary units) of pyramidal neurons in layer 2/3 and layer five for a total of 200 pyramidal neurons, with interneurons interleaved regularly in a 3–1 ratio (see *Figure 3D*). The local synaptic architecture in *Figure 3A* was based on an abundance of animal studies and, in particular, studies of the mouse/rat somatosensory cortex (*Bernardo et al., 1990a*; *Bernardo et al., 1990b*; for review, see *Thomson et al., 2002*; *Thomson and Bannister, 2003*, and *Bannister, 2005*). Inhibitory synaptic connections onto PNs were located on the soma (*Freund et al., 1986*; *Kisvárday et al., 1985*; *Somogyi et al., 1983*), and excitatory synapses contacted the basal and apical oblique dendrites (*Deuchars et al., 1994*; *Feldmeyer et al., 2002*; *Lübke et al., 1996*; *Thomson and Bannister, 1998*).

Synaptic dynamics were modeled with bi-exponential functions. The rise and decay time constants and reversal potentials were based on experiments and the original neocortical model in *Jones et al. (2009)*, and are generally as follows: AMPA (0.5 ms, 1.0 ms, 0 mV); NMDA (1.0 ms, 20.0 ms, 0 mV); GABAA (0.5 ms, 5.0 ms, −80 mV), GABAB (1.0 ms, 20.0 ms, −80 mV). Within a cortical layer there is recurrent connectivity between neurons of a given type (PN to PN, interneuron to interneuron), PN to interneuron connectivity, and synaptic inhibition from interneurons onto PNs. The following synaptic connections are present across cortical layers: layer 2/3 PNs to layer 5 PNs, layer 2/3 interneurons to layer 5 PNs, layer 2/3 PNs to layer five interneurons.

The conductance of the synaptic connections within the local network grid were defined with a symmetric 2D Gaussian spatial profile, with a delay incorporate into the synaptic connection between two cells defined by and inverse Gaussian (*Jones, 1986*; *Kaas and Garraghty, 1991*). There is all-to-all connectivity between any two populations of synaptically-coupled neurons. Synaptic *weights* between the neurons are scaled inversely by the distance in the XY plane (arbitrary units) between the neurons ($d$) using exponential fall-off following $e^{-d^2/\lambda^2}$, and space constant $\lambda$, which depends on pre- and post-synaptic type (*Table 2* below).

The synaptic *delays* are scaled in proportion to the XY plane distance ($d$) between the neurons following $1/e^{-d^2/\lambda^2}$, to account for the larger propagation distance (note that the $\lambda$ value is determined using values in *Table 2*). With increasing $d$ between neurons, the synaptic weights decay, while the

**Table 2.** Space constant (arbitrary units) for synaptic connection strengths and delays between different populations of neurons (rows are pre-synaptic type, columns are post-synaptic type).

| (From ↓, To →) | L2/3 PN | L2/3 Basket | L5 pn | L5 basket |
|---|---|---|---|---|
| L2/3 PN | 3 | 3 | 3 | 3 |
| L2/3 Basket | 50 | 20 | 50 | N/A |
| L5 PN | 3 | N/A | 3 | 3 |
| L5 Basket | N/A | N/A | 70 | 20 |

synaptic delays increase. The connectivity details are based on known neocortical anatomy and local circuit wiring patterns, as derived from the literature. Further details on connectivity are available on HNN's website and prior publications.

## Exogenous driving inputs

At rest, the default model does not generate activity. HNN provides several ways to activate the local cortical column with layer specific excitatory synaptic input representing thalamo-cortical, and/or cortical-cortical and noisy/tonic drive. The user defines the choice of driving input to the network, based on their simulation experiment, as described in Results.

Exogenous driving networks are not explicitly modeled, rather the user defines trains or bursts of action potentials representing these inputs that excite the local network via AMPA or NMDA synaptic connections to distinct layers and cellular compartments. These inputs are referred to as proximal and distal drive based on the PN dendritic contact location. Proximal inputs contact basal and oblique dendrites of PN and somas of the inhibitory neurons in L2/3 and L5, and distal inputs contact distal dendrites of the PN in L2/3 and L5 and somas of the inhibitory neurons in L2/3 only, as shown in *Figure 3*.

The trains of action potentials, or tonic/noisy input, that the user defines are created in specific dialog boxes in the GUI and represent either Evoked, Rhythmic, Tonic, or Poisson Inputs, as motivated by our prior studies and tutorials described in Results.

### Evoked input

Evoked inputs are trains of synaptic inputs to the local network during a sensory stimulus that creates an event related potential (ERP). Parameter choices for defining these inputs are shown in *Figure 4A*. The following parameter values are used to define each proximal or distal evoked input:

- Start time mean (ms) - average start time
- Start time stdev (ms) - standard deviation of start time
- Number spikes - number of inputs provided to each synapse
- L2/3 Pyr weight AMPA/NMDA ($\mu S$) - weight of AMPA/NMDA synaptic inputs to layer 2/3 pyramidal neurons
- L2/3 Basket weight AMPA/NMDA ($\mu S$) - weight of AMPA/NMDA synaptic inputs to layer /32 basket cells
- L5 Pyr weight AMPA/NMDA ($\mu S$) - weight of AMPA/NMDA synaptic inputs to layer 5 pyramidal neurons
- L5 Basket weight AMPA/NMDA ($\mu S$) - weight of AMPA/NMDA synaptic inputs to layer 5 basket cells (only used for *proximal* inputs)

Each evoked input also has a 'Synchronous Inputs' option, indicating whether for a specific evoked proximal/distal input each neuron receives the input at the same time, or if instead each neuron receives the evoked input events independently drawn from the same distribution. Increment input (ms) indicates whether to increment the Start time of all evoked inputs on each trial. In the studies described above, the evoked input strengths are suprathreshold generating action potentials in the local network.

### Rhythmic input

Rhythmic Inputs are typically bursts of action potentials that drive the local network rhythmically. Parameter choices for defining these inputs are shown in *Figure 7A*. Each rhythmic input is defined

as a series of 'population bursts', consisting of a set number of 'burst units' which drive post-synaptic conductances in the local network with a set frequency and mean delay between proximal and distal projections. Rhythmic proximal and distal inputs target different cortical layers, as described above. HNN allows setting proximal and distal rhythmic synaptic input start/stop times and frequencies using the following specification:

- Start time mean (ms) - specifies the average start time for rhythmic inputs
- Start time stdev (ms) - specifies the standard deviation of start times for rhythmic inputs
- Stop time (ms) - specifies when the rhythmic inputs should be turned off
- Burst frequency (Hz) - average frequency of bursts
- Burst stdev (ms) - standard deviation of input events
- Spikes/burst - provides *n* synaptic events at each selected time
- Number bursts - number of times the full Burst sequence is repeated (each repeat adds variability and more inputs)

In addition, HNN's Rhythmic Input dialog box allows setting the weights of the rhythmic synaptic inputs (units of conductance) to individual neuron types in layers 2/3 and 5, and adding synaptic delays (ms) before the neurons receive the synaptic inputs. In the studies described above, rhythmic inputs are set to sub-threshold synaptic strengths, and therefore do not lead to neuronal action potentials.

## Tonic/Noisy Input

Tonic inputs are modeled as somatic current clamps with a fixed current amplitude (nA). These clamps can be used to adjust the resting membrane potential of a neuron, and bring it closer (with positive amplitude injection) or further from firing threshold (with a negative amplitude injection). Parameter choices for defining these inputs are shown in *Figure 9A* and include setting the current clamp amplitude, and start/stop time for each modeled neuron type separately.

Noisy Inputs are trains of action potentials that follow a Poisson Process and create excitatory AMPA or NMDA synaptic inputs to the somata of all neurons of a given type. Parameter choices for defining these inputs are shown in *Figure 9A* and include, setting the average frequency of the Poisson drive, synaptic strength to somatic AMPA or NMDA synapses, and start/stop times of all Poisson inputs.

## Calculation of primary electrical current (Net Current Dipole)

Axial current flow between any two neighboring model compartments i,j is defined as $i_{axial} = (v_i - v_j)/r_{axial}$, where $v_i$, $v_j$, and $r_{axial}$ are the voltages in compartment i, j, and the resistance between the compartments, respectively. In order to convert this axial current into a dipole signal, we apply a length scaling where the axial current is scaled by the inter-compartment distance along the vertical axis. The length scaling means that for the longer apical dendrites of layer five pyramidal neurons, the contribution will be larger than from the shorter layer 2/3 pyramidal neuron apical dendrites. Note that the orientation of the dendrites relative to the vertical axis also influences the contribution to the dipole signal. For example, the horizontally-oriented oblique dendrites which do not have any vertical length component, do not contribute to the dipole signal, whereas for basal dendrites oriented at 45 degrees from the vertical axis, the scaling is $-\sqrt{2}/2$ (note the negative sign is because these dendrites are pointing downward). The contribution from all neighboring compartments within a neuron is integrated and then added to a value across the set of all pyramidal neurons. As a result of the multiplication between axial current and length, the model dipole output signal has the same units of measure as the experimental data (Am) (*Okada et al., 1997*; *Murakami et al., 2003*; *Murakami and Okada, 2006*; *Jones et al., 2007*; *Hagen et al., 2018*).

## ERP optimization tools

HNN includes a method to optimize ERP simulations. The optimization procedure was uniquely designed to minimize the RMSE between model output and ERP waveforms in a stepwise manner that decreases parameter exploration and saves compute time. This procedure takes advantage of the assumption that the exogenous proximal and distal driving inputs are essential parameters to tune to get an accurate representation of an ERP waveform. Additionally, it applies the knowledge

that, with probabilistic certainty, features of the dipole waveform at a particular point in time cannot be influenced by an exogenous driving input that begins after that point in time.

Since exogenous inputs are modeled as Gaussian processes, the likelihood of occurrence can be modeled by a probability distribution function (PDF) normally distributed with a given mean and standard deviation. *Figure 11—figure supplement 1A* shows the PDFs of the inputs for the supra-threshold example described in the results *Figure 11*. An input's contribution to the ERP will begin when there is a non-zero probability of occurrence and persist for a duration commensurate with the input's cumulative distribution function (CDF), shown in *Figure 11—figure supplement 1B*. This clearly illustrates that from 20 to 50 ms, the input labeled 'Proximal 1' is the unique contributor to the waveform. After 50 ms, effects from Distal one begin, thus adding new parameters that contribute to the waveform fit and reduce the relative contribution of Proximal 1 (from full to partial). It follows that each successive driving input will have a time window where it is most likely to have a unique and dominant effect. As such, our approach to model optimization is to divide the process into smaller steps where only a single input's parameters are estimated before proceeding to optimize the next input.

To implement this procedure, we developed a new goodness of fit measure that amplifies the importance of maximizing the fit at points of unique contribution (e.g. 20–50 ms for Proximal 1, *Figure 11—figure supplement 1C*) and diminished the importance of fitting to later points where other inputs contribute more to the fit. We began with standard root mean squared error (RMSE)

$$\text{RMSE} = \sqrt{\frac{\sum_{t=0}^{T} (x_{1,t} - x_{2,t})^2}{T}}$$

where $t$ is the current simulation time, from 0 to simulation completion ($T$), and $x_{1,t}$ is the simulated dipole at $t$, and $x_{2,t}$ is the experimental data point. Then we adapted RMSE to include weight functions specific for input $k$ at time $t$,

$$\text{wRMSE}_k = \sqrt{\frac{\sum_{t=0}^{T} w_k(t)(x_{1,t} - x_{2,t})^2}{\sum_{t=0}^{T} w_k(t)}}$$

where an assignment of $w_k(t)=1$ for all $t$ would be equivalent to RMSE.

For each input $k$, we first defined a weight distribution function, $w_k(t)$, as the Unique Contribution Index (UCI), which starts from the CDF of input $k$ and simply subtracts the CDF of subsequent inputs, with a lower bound of 0 (*Figure 11—figure supplement 1C*). Equivalently,

$$\text{UCI}_k(t) = \text{CDF}_k(t) - \sum_{i=k+1}^{N} \text{CDF}_i(t),$$

where $N$ is the number of exogenous driving inputs in the simulation. *Figure 11—figure supplement 1C* shows that Proximal 1's influence is unique up to 50 ms, Distal one has a dominant, but not unique contribution near 70 ms, and Proximal two is dominant after ~100 ms. When the UCI is applied as a weighting function in the wRMSE equation above, we observed that some optimization steps would negatively impact the fit in regions after the peak in UCI, where the errors had been down-weighted, requiring subsequent optimization steps to attempt to 'correct' the fit. Our solution was to instead define the weight function using the Extended Contribution Index (ECI), which includes a term that delays the weight function's return to 0, extending the window of data points that have an impact on wRMSE further into the simulation. This achieves a balance between optimal parameter estimates for the current step and providing a good starting point for following optimization steps. ECI is defined by

$$\text{ECI}_k(t) = \text{CDF}_k(t) - \sum_{i=k+1}^{N} \text{CDF}_i(t) \, A\left(\frac{\mu_i - \mu_k}{T}\right),$$

where $\mu_i$ and $\mu_k$ are the mean start times of the next input and the current input, respectively.

Simulation length is represented by $T$ and $A$ is an empirically derived constant. We arrived at a value of 1.6 for $A$ as a factor that appropriately minimized the contribution of inputs proportional to the delay between their onset and the $k^{\text{th}}$ input currently being optimized. The effect of the ECI's decay term can be seen in *Figure 11—figure supplement 1D* where the ECI for Proximal one extends further than the corresponding UCI, and the ECI of Distal one remains significant through the end of the simulation. Since points where $ECI_{k,t}$ approximately equal 0 will have a negligible impact on $wRMSE_k$, we define a threshold of 0.01 where $wRMSE_k$ is calculated for the window starting when $ECI_{k,t}$ rises above 0.01 and ending when $ECI_{k,t}$ drops below 0.01. For the first exogenous driving input, it is likely that the window will end before the completion of the simulation. In that first step, simulations can be stopped early, reducing the time required for simulating each candidate parameter set in that step.

The final step in our model optimization process is to vary all free parameters from all inputs using regular RMSE to measure goodness of fit. Like each previous step, the number of simulations run is limited. So this primary purpose of this final step is to make small corrections, not perform all-at-once optimization (which would likely require thousands of simulations). It also provides an opportunity to rebalance the contributions from multiple inputs in regions where there is a high degree of parameter inter-dependence. However, if the user is certain that they want to perform all-at-once optimization (which would likely require many more simulations), they could set the number of simulations for all steps except the last one to 0, and specify a very large number of simulations for the final step.

For each optimization step, HNN uses the COBYLA optimization algorithm (*Powell, 1994*), which supports bound constraints as defined by the user for each parameter. We have found COBYLA converges at a local minimum faster than the PRAXIS algorithm (*Brent, 1973*) as implemented in NEU-RON's multiple run fitter.

## Acknowledgements

Research supported by NIH NIBIB BRAIN Award 5-R01-EB022889-02, NIH NIBIB BRAIN Award Supplement R01EB022889-02S1, NIH NIDCD 5-R01DC012947-07, ARO W911NF-19-1-0402. We thank Dimitrios Pantazis and colleagues for supplying source localized data for our gamma tutorial.

## Additional information

### Funding

| Funder | Grant reference number | Author |
| --- | --- | --- |
| National Institute of Biomedical Imaging and Bioengineering | BRAIN Award 5-R01-EB022889-02 | Samuel A Neymotin<br>Dylan S Daniels<br>Blake Caldwell<br>Robert A McDougal<br>Nicholas T Carnevale<br>Mainak Jas<br>Christopher I Moore<br>Michael L Hines<br>Matti Hämäläinen<br>Stephanie R Jones |
| National Institute of Biomedical Imaging and Bioengineering | BRAIN Award Supplement R01EB022889-02S1 | Samuel A Neymotin<br>Dylan S Daniels<br>Blake Caldwell<br>Robert A McDougal<br>Nicholas T Carnevale<br>Mainak Jas<br>Christopher I Moore<br>Michael L Hines<br>Matti Hämäläinen<br>Stephanie R Jones |
| National Institute on Deafness and Other Communication Disorders | 5-R01DC012947-07 | Samuel A Neymotin |
| Army Research Office | W911NF-19-1-0402 | Samuel A Neymotin |

The funders had no role in study design, data collection and interpretation, or the decision to submit the work for publication. The views and conclusions contained in this document are those of the authors and should not be interpreted as representing the official policies, either expressed or implied, of the Army Research Office or the U.S. Government. The U.S. Government is authorized to reproduce and distribute reprints for Government purposes notwithstanding any copyright notation herein.

### Author contributions
Samuel A Neymotin, Dylan S Daniels, Blake Caldwell, Robert A McDougal, Nicholas T Carnevale, Mainak Jas, Christopher I Moore, Conceptualization, Resources, Data curation, Software, Formal analysis, Funding acquisition, Validation, Investigation, Visualization, Methodology, Writing - original draft, Writing - review and editing; Michael L Hines, Matti Hämäläinen, Stephanie R Jones, Conceptualization, Resources, Data curation, Software, Formal analysis, Supervision, Funding acquisition, Validation, Investigation, Visualization, Methodology, Writing - original draft, Project administration, Writing - review and editing

### Author ORCIDs
Samuel A Neymotin  https://orcid.org/0000-0003-3646-5195
Blake Caldwell  https://orcid.org/0000-0002-6882-6998
Robert A McDougal  http://orcid.org/0000-0001-6394-3127
Christopher I Moore  http://orcid.org/0000-0003-4534-1602
Stephanie R Jones  https://orcid.org/0000-0001-6760-5301

### Decision letter and Author response
Decision letter https://doi.org/10.7554/eLife.51214.sa1
Author response https://doi.org/10.7554/eLife.51214.sa2

## Additional files

### Supplementary files
• Supplementary file 1. Summary of weighted total Sobol sensitivity index values averaged across the entire simulation for each exogenous driving input in two sensory evoked response models: suprathreshold (*Figure 11*) and 50% detected (*Figure 4*) . The ranking of each input's parameters is similar between models.

• Transparent reporting form

### Data availability
All source-code, model parameters, and associated data are provided in a permanent public-accessible repository on github (https://github.com/jonescompneurolab/hnn).

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

## Appendix 1

### Supplementary materials

#### Sensitivity analyses of ERP simulations

To reduce the computational demands of performing model optimization of HNN ERP simulations, we used variance based sensitivity analysis to identify parameters that were less significant to the simulated dipole waveform. As discussed above, HNN's model optimization feature focuses on estimating parameters of the exogenous driving inputs. Of those parameters, ones that did not vary model output significantly, as determined by sensitivity analysis, could be excluded from parameter estimation.

The method of variance based sensitivity analysis through Monte Carlo estimation (*Sobol', 2001*) provides Sobol sensitivity indices that can be used to explain the relative contribution of individual parameters on model variance. The total Sobol sensitivity index for each parameter serves as a measure that represents that parameter's contribution to the variance, and also the contributions resulting from interactions with other parameters being varied (*Homma and Saltelli, 1996*). So a parameter with a low total Sobol sensitivity index can be characterized as an insignificant contributor to variance and can be fixed at its default value during model optimization.

We used the Python package Uncertainpy (*Tennøe et al., 2018*) to perform sensitivity analyses of parameters belonging to the exogenous driving inputs in the perceptual threshold-level ('yes_trial_SI_ERP_all_avg.txt') and suprathreshold-level ('ERPYesSupraT.txt') evoked response examples provided with HNN and described in the Results (*Figure 4* and *Figure 11*). These analyses were performed using a modified simulation interface to run the simulations in parallel on a high-performance computing cluster, which is not currently included with HNN distribution. The results from our sensitivity analyses are shown in *Figure 4—figure supplement 1*, *Figure 11—figure supplement 2*, and *Figure 11—figure supplement 1*. Each analysis consisted of varying all parameters (except input timing standard deviation) of a driving input over 55,000 simulations using a quasi-Monte Carlo method that sampled from parameter distributions we specified. The input time distribution was defined as a normal distribution with the same mean and standard deviation in the corresponding HNN parameter file. The values for various synaptic weights were chosen from a uniform distribution ranging from the default value plus or minus 500%. For synaptic weight parameters with a default value of 0, the uniform distribution ranged from 0 to 1.0.

We were interested in comparing the contribution of each parameter to the dipole waveform (in units of nAm) across the entire simulation time. However, since the calculation of the total Sobol sensitivity index is relative to the variance at each point of the time series, when variance changes over time, it is not appropriate to average the total Sobol sensitivity indices across the entire simulation (*Alexanderian et al., 2020*). Instead we computed a weighted total Sobol index at each point in time (weighted by std. deviation scaled to have a range from 0 to 1). The plots in *Figure 4—figure supplement 1* and *Figure 11—figure supplement 2* show weighted total Sobol indices for each parameter over the duration of the simulation. *Supplementary file 1* ranks the parameters with the greatest contribution to model output using the arithmetic means of weighted total Sobol indices across the entire simulation, for each driving input.

The results from our sensitivity analyses of sensory evoked response examples illustrate that there are several candidate parameters for excluding from model optimization. Not surprisingly input timing is an important parameter to optimize. We also found that NMDA weights typically have a greater contribution to variance than AMPA weights, as do connections to Layer 5 neurons compared to Layer 2/3 neurons.

