## [Decision Letter]

**Acceptance summary:**

The HNN provides an intuitive way for thinking about and linking cellular- and circuit-level mechanisms in the neocortex to (source-reconstructed) neural data. Combined with the worked examples that demonstrate the application of the tool to current issues, the reviewers unanimously agreed on its relevance and potential for opening up new possibilities for the field to advance our understanding of cellular and network origins of MEG/EEG data.

**Decision letter after peer review:**

Thank you for submitting your article "Human Neocortical Neurosolver (HNN), a new software tool for interpreting the cellular and network origin of human MEG/EEG data" for consideration by *eLife*. Your article has been reviewed by three peer reviewers, including Arjen Stolk as the Reviewing Editor and Reviewer #1, and the evaluation has been overseen by Richard Ivry as the Senior Editor. The following individual involved in review of your submission has agreed to reveal their identity: Sarang S. Dalal (Reviewer #2).

The reviewers have discussed the reviews with one another and the Reviewing Editor has drafted this decision to help you prepare a revised submission.

Summary:

All reviewers and editors agreed on the methodological sophistication and relevance of your software tool. However, several concerns were raised, which are summarized below. We are aware that these may be challenging to address within the usual timeframe but we all agreed that if these issues can be addressed satisfactorily, the paper would substantially improve.

Essential revisions:

1) MEG is generally considered to arise from intracellular currents, with EEG arising from extracellular currents (e.g., section 2.1 https://www.sciencedirect.com/science/article/pii/S0896627313009203). In the case of MEG, the volume currents largely cancel out, which is thought to leave intracellular currents as the main contributor to MEG signals. But volume currents are supposed to be a dominating factor in EEG, so this theoretically leads to its bias towards extracellular currents. However, the manuscript and software appear to address only intracellular currents, while MEG and EEG are discussed as if they are equivalent. This may be a necessary compromise, or the authors may disagree with the premise of MEG from intracellular and EEG from extracellular currents. However, they should be clear on their position and how this issue is handled in HNN.

2) It is a bit unsatisfying that the "Gamma rhythms" example does not use real data, given its importance to the community. Even if the authors' own experiments have not yielded data with strong gamma, there are many examples in the literature that could perhaps be obtained, and even some appropriate datasets are publicly available. For example, FieldTrip's gamma source localization tutorial provides one:

http://www.fieldtriptoolbox.org/tutorial/beamformingextended/

3) The HNN framework seems well-suited for evoked responses that are phase-locked across trials. However, gamma oscillations are often "induced", i.e., not phase-locked across trials. Does HNN provide tools to model/simulate the origins of such phase variance? If so, this should be clarified, and if not, perhaps described as a limitation.

4) There appears to be a distinction between narrowband and broadband gamma in terms of their neural underpinnings, with broadband gamma suggested to be closely linked to spiking activity (e.g. Whittingstall and Logothetis, 2009 among others). Likewise, this should be described if HNN can model this, and otherwise addressed as a limitation.

5) The "jet" colormaps for the spectrograms (Figures 6 and 7) have been heavily criticized in recent years for being extremely biased perceptually (see, e.g., https://predictablynoisy.com/makeitpop-intro). In the case of Figure 6, this results in yellow patches that "pop" somewhat misleadingly. New releases of neuroscientific software should be especially cognizant of this and ideally provide a perceptually uniform colormap as the default. See https://matplotlib.org/3.1.1/tutorials/colors/colormaps.html and https://matplotlib.org/cmocean/ for some ideas on divergent colormaps.

6) The manuscript refers to previous work by the authors for details of the biophysical model. However, it is not clear from the present manuscript how well these models compare to actual experimental data and from which species the data was obtained. For example, because the model appears to use reduced and simplified neuronal morphologies, and only a single type of inhibitory interneuron, it would be very informative to provide some assessment of whether any potential inaccuracies in the generated field potentials could exist due to reduced dendritic morphology, and whether incorporation of additional inhibitory circuitry could expand the scope of questions that could be addressed via HNN.

7) The manuscript refers several times to source localization, but the presented model appears to be of a single cortical column, and presumably, this means that it is currently not possible to study questions related to the spatial distribution of field potentials. While I have no doubt that the methods of HNN will scale well to networks of multiple cortical columns, some clarity is necessary in manuscript regarding the current capability of HNN with regards to spatially distributed networks and source localization features, and what plans are made for future releases.

8) While the manuscript addresses related work by other authors, there are now several published large-scale biophysical neural models that incorporate some form of approximated LFP computation that aids the study of neural oscillations. It would be very informative to include at least a brief comparison with e.g. the methods developed by Reimann et al., 2013, and in what ways HNN is an improvement.

---

## [Author Response]

Essential revisions:1) MEG is generally considered to arise from intracellular currents, with EEG arising from extracellular currents (e.g., section 2.1 https://www.sciencedirect.com/science/article/pii/S0896627313009203). In the case of MEG, the volume currents largely cancel out, which is thought to leave intracellular currents as the main contributor to MEG signals. But volume currents are supposed to be a dominating factor in EEG, so this theoretically leads to its bias towards extracellular currents. However, the manuscript and software appear to address only intracellular currents, while MEG and EEG are discussed as if they are equivalent. This may be a necessary compromise, or the authors may disagree with the premise of MEG from intracellular and EEG from extracellular currents. However, they should be clear on their position and how this issue is handled in HNN.

Thank you for bringing up the need to address this basic conceptual point in more detail. As even the reference to a publication by a very influential author in the field demonstrates, the biophysics of the generation and the concepts used in the modeling of MEG/EEG need to be further clarified and the correct physically sound concepts have to be restated. There is much confusion in the field on this topic.

We, indeed, disagree with the notion that MEG is generated by the intracellular and EEG with the extracellular currents. In fact, in the modeling of the MEG and EEG signals at a long distance of at least centimeters from the actual sources, the concepts of “intracellular” and “extracellular” do not directly apply. Instead, a macroscopic scale is assumed for the distribution of electrical conductivity. The proper division between the actual non-ohmic equivalent sources of activity and the passive ohmic currents is then referred to as primary and volume currents, respectively. Both MEG and EEG are ultimately generated by the primary currents, which sets up a potential distribution (EEG) whose gradients are driving the volume currents. Both primary and volume currents, in general, create the magnetic field (MEG). Surprisingly, the conductivity distribution and geometry of the compartments with different electric conductivities in the head leads to a situation in which the integral effect of the volume currents can be relatively easily taken into account while the precise computation of the EEG for a given primary current requires more detailed information about the electrical conductivity. We want to point out that these concepts are succinctly discussed in [1] and [2].

Given the primary currents and the macroscopic distribution of electrical conductivity, the computation of MEG/EEG can be accomplished by using either an analytical method (in special cases) or by employing numerical boundary-element, finite-element, or finite-difference methods. Source estimation methods aim at finding the best estimate for the primary currents underlying the MEG/EEG measurements.

The distinct purpose of HNN is to interpret how the primary current distribution corresponding to the microscopic currents is created by an assembly of neurons. Thanks to the consistent orientation of the apical dendrites of the pyramidal cells in the cortex this primary current is oriented normal to the cortical mantle and its direction corresponds to the intracellular current flow [3, 4, 5]. By definition, the same primary current is the source of both MEG and EEG.

The clarify this in the manuscript, we have rewritten the Results subsection titled

“Primary currents and the relation to forward and inverse modeling”. In addition, we have created a new panel in Figure 1B that emphasizes the relationship between the primary currents, MEG/EEG forward and inverse modeling and HNN. We hope that these additions to the manuscript bring further clarity to the confusing topic. However, we also emphasize that the goal of the current HNN methods paper is not to review electromagnetic biophysics, but rather to present a new tool for neural interpretation of the primary currents underlying MEG/EEG signals.

2) It is a bit unsatisfying that the "Gamma rhythms" example does not use real data, given its importance to the community. Even if the authors' own experiments have not yielded data with strong gamma, there are many examples in the literature that could perhaps be obtained, and even some appropriate datasets are publicly available. For example, FieldTrip's gamma source localization tutorial provides one:http://www.fieldtriptoolbox.org/tutorial/beamformingextended/

As a result of the reviewer’s insightful suggestion, we have obtained source-localized current dipole signals from human MEG recordings showing visually-induced gamma rhythms. This data set has been published (Pantazis et al., 2018) and was generously provided by the authors. In these data, induced gamma rhythms are not continuous but emerge as transient bursts of activity on individual trials. We were able to model this type of gamma activity using HNN with a slight modification, the weak PING model in our previous gamma tutorial (Figure 9). Specifically, we adjusted the properties of the noisy drive to the network. HNN modeling of this new data and a corresponding new figure (Figure 10) are now described in the “Gamma rhythms” subsection of the manuscript section “Tutorials on ERPs and low-frequency oscillations”. We also uploaded the experimental data to the HNN github repository. As stated at the end of the gamma tutorial section: “This example is presented as a proof of principle on the method to begin to study the cell and circuit origin source localized gamma band data with HNN. The circuit level hypotheses have not yet been published elsewhere or investigated in further detail.”

Of note to the reviewer, the gamma source localization data distributed with FieldTrip is not source localized and as such requires additional processing before it can be compared to the output in HNN. Therefore, we decided to use the alternative source localized data set provided by Pantazis et al., 2018, as described. This data set was already in units of nAm and could be directly compared to the HNN model output in both the time and frequency domain without further processing (Figure 10).

3) The HNN framework seems well-suited for evoked responses that are phase-locked across trials. However, gamma oscillations are often "induced", i.e., not phase-locked across trials. Does HNN provide tools to model/simulate the origins of such phase variance? If so, this should be clarified, and if not, perhaps described as a limitation.

We thank the reviewer for pointing out this lack of clarity. Indeed, non time-locked spontaneous and “induced” oscillations are fundamental brain dynamics that HNN can be used to understand. We have addressed this comment in two ways.

First, we note that in reply to comment #2 above, we have added a new tutorial and section in the methods paper that describes how to simulate induced gamma oscillations (see above, and new Figure 1).

Second, our prior studies of spontaneous alpha/beta activity described in our alpha and beta tutorial focused on non time-locked rhythms. This dataset is provided and described as “induced” in the “Getting Started” section of the Alpha/Beta tutorial. We also point out in the tutorial “Section 5” that due to the non-timelocked nature of the signal, it is not possible to directly compare the time domain output of the model with single epoch time traces. However, data can be directly compared to model in the Power Spectral Density viewer. Additionally, tutorial “Section 3.3” describes how to create variability in the drive to the network on different trials that effectively jitters the phase of the induced oscillations on each trial. In the averaged response, seemingly continuous high power activity can occur.

We have now clarified this in the tutorials by updating the title of the tutorial sections to emphasize the induced nature of the data we are simulating.

“Section 3.3: Simulating multiple trials of non-time locked spontaneous or induced signals with jitter, and averaging across trials, creates the impression of continuous oscillations”

“Section 5: Comparing model output and recorded data for non-timelocked spontaneous or induced oscillations”

We have also clarified in several places in the text describing alpha and beta rhythms, and gamma rhythms, that these simulations are constructed to study non-time locked spontaneous or “induced” rhythms.

4) There appears to be a distinction between narrowband and broadband gamma in terms of their neural underpinnings, with broadband gamma suggested to be closely linked to spiking activity (e.g. Whittingstall and Logothetis, 2009 among others). Likewise, this should be described if HNN can model this, and otherwise addressed as a limitation.

Indeed, there is much literature addressing the neural underpinnings of narrowband vs. broadband gamma oscillations, and we have used HNN to contribute to this literature (Lee and Jones, 2013). In the methods paper and in our online tutorials, we have chosen to only describe application of HNN to narrowband low gamma PING mechanism.

To directly address this concern, we have now explicitly stated in the methods paper that HNN has been applied to study the neural underpinnings of broadband “high gamma” activity with reference to Lee and Jones 2013, where these issues are outlined further. Specifically, the following statements have been added to the paper:

“It has been well established through experiments and computational modeling that these rhythms can emerge in local spiking networks through excitatory and inhibitory cell interactions where synaptic time constants set the frequency of the oscillation, while broadband or “high gamma” rhythms reflect spiking activity in the network that creates sharp waveform deflections (Lee and Jones, 2013). “

“We have applied HNN to determine if features in the current dipole signal could distinguish PING-mediated gamma from other possible mechanisms such as exogenous rhythmic drive or spiking activity that creates “high gamma” oscillations (Lee and Jones, 2013). Here, we describe the process for generating gamma rhythms via the canonical PING mechanisms in HNN.”

In response to critique 2, we have also added an additional example of simulating gamma and comparing to recorded data to the methods paper (Figure 10). The new data and parameter files are now included with the HNN installation so that others can replicate the new simulation.

5) The "jet" colormaps for the spectrograms (Figures 6 and 7) have been heavily criticized in recent years for being extremely biased perceptually (see, e.g., https://predictablynoisy.com/makeitpop-intro). In the case of Figure 6, this results in yellow patches that "pop" somewhat misleadingly. New releases of neuroscientific software should be especially cognizant of this and ideally provide a perceptually uniform colormap as the default. See https://matplotlib.org/3.1.1/tutorials/colors/colormaps.html and https://matplotlib.org/cmocean/ for some ideas on divergent colormaps.

We thank the reviewer for pointing out this criticism in the field. To directly address this concern, we have updated HNN so that the user has a choice of colormap for their spectrogram images. This is described in the paper in the section “Alpha and beta rhythms” section under “Tutorials on ERPs and low-frequency oscillations”, along with panels in Figure 6 (Figure 6B and C).

Due to the fact that our prior publications that serve as the basis for the existing tutorials have used the “jet” colormap, we have chosen not to change the figures in our tutorials for consistency and to avoid adding confusion.

6) The manuscript refers to previous work by the authors for details of the biophysical model. However, it is not clear from the present manuscript how well these models compare to actual experimental data and from which species the data was obtained. For example, because the model appears to use reduced and simplified neuronal morphologies, and only a single type of inhibitory interneuron, it would be very informative to provide some assessment of whether any potential inaccuracies in the generated field potentials could exist due to reduced dendritic morphology, and whether incorporation of additional inhibitory circuitry could expand the scope of questions that could be addressed via HNN.

The reviewer is correct that the underlying neural model is a reduced representation of neural activity. The initial choices made in the reduction were based on including the minimal components necessary to account for the primary current dipole signal (from pyramidal neuron dendrites) and to maintain canonical features of neocortical circuitry (layers, E/I ratio, synaptic time-constants, etc.). We’ve found that this reduction has been sufficient to account for important features of the most commonly measured macroscale signals (ERPs and low-frequency oscillations) and to generate new, testable hypotheses on the origin of these signals (e.g. beta bursts; Sherman et al., 2016).

We agree that this reduction may cause for some inaccuracies in simulating extracellular local field potential (LFP) signals that may rely on more detail in the neocortical circuit and/or on different interneuron types. This is yet to be determined and simulating LFP is not currently part of HNN. As discussed, we are interested in expanding HNN to be useful to simulate LFPs, and this is an ongoing effort. As a first step, we are converting the code responsible for building the NEURON model to NetPyNE so that other neuron types, such as different types of pyramidal cells or interneurons, can be implemented into the model. This expansion may also reveal the importance of yet-to-be-implemented features of the neural circuit in simulating the primary current signal. Our vision is for HNN, as open-source software, to be adaptable, so that new information can easily be incorporated into HNN and shared with the community.

To directly address the reviewers concern, we have added further details in the Template Model Construction in the Materials and methods sections: “Morphology”, “Physiology”, and “Local Network Connections”. These sections now detail the choices made in cell and network construction and the publications that motivated our original model. Of note, we did not include additional equations or parameters, as all equations are part of the open-source code.

Additionally, we have expanded our Discussion section to further address limitations in the model. The “Limitation and future directions” section of the Discussion reads:

“One of the greatest challenges in computational neural modeling is deciding the appropriate scale of model to use to answer the question at hand. […] These assumptions are outlined in detail in the Materials and methods section, in our prior publications, and on our website.”

7) The manuscript refers several times to source localization, but the presented model appears to be of a single cortical column, and presumably, this means that it is currently not possible to study questions related to the spatial distribution of field potentials. While I have no doubt that the methods of HNN will scale well to networks of multiple cortical columns, some clarity is necessary in manuscript regarding the current capability of HNN with regards to spatially distributed networks and source localization features, and what plans are made for future releases.

We hope that the response to comment #1 has helped clarify this issue. Additionally, we have described in several places throughout the manuscript that HNN is designed to simulate the activity in a single source region of interest, and that future expansions will include simulation of spatially distributed networks and sensor level signals. Example clarifications are as follows.

In the subsection, “Inferring the neural origin of the primary currents with HNN**”**

“Currently, the process of estimating the primary current sources with inverse methods, or calculating the forward solution from **J**^p^to the measured sensor level signal, is separate from HNN.”

“HNN is currently constructed to dissect the cell and network contributions to signals from one source-localized region of interest. […] Ongoing expansions will include the ability to import other user-defined cell types and circuit models into HNN, simulate LFP and sensor level signals, as well as the interactions among multiple neocortical areas (see Discussion).”

In the subsection, “Comparison to other neocortical models and EEG/MEG modeling software”:

**“**This construction enables one-to-one comparison between model output and source localized data in units of Am, and was specifically designed for interpreting microscale cellular- and circuit-level activity from single regions of interest.”

“The LFPy Python classes are likely to provide a useful framework for expanding the utility of HNN to include multi-area simulations, and simulations of LFP and EEG/MEG sensor level signals, as described in Limitations and future directions below. “

In the subsection “Limitations and future directions”

**“**HNN’s model was chosen to be minimally sufficient to accurately account for the biophysical origin of the primary currents that underlie EEG/MEG signals in a single brain area; namely, the net intracellular current flow in the apical dendrites of pyramidal neurons that span across the cortical layers and receive layer-specific synaptic input from other brain areas. […] A future direction discussed below is to expand HNN to simulate extracellular local field potential signals (LFPs), and sensor level signals, whose accuracy may require additional model detail and whose implementation can be aided by other existing tools such as LFPy and NetPyNe (discussed further below)”

8) While the manuscript addresses related work by other authors, there are now several published large-scale biophysical neural models that incorporate some form of approximated LFP computation that aids the study of neural oscillations. It would be very informative to include at least a brief comparison with e.g. the methods developed by Reimann et al., 2013, and in what ways HNN is an improvement.

We thank the reviewer for pointing out that we were not clear in describing how our model was different from existing models with similar capabilities. Indeed there are many large-scale biophysical neural models to simulate various aspects of neocortical activity, including LFP, and source localized and EEG/MEG sensor level signals. Several modeling tools and associated documentation are also available to build user defined neocortical models for general use that is not domain specific. Among the current modeling software designed specifically for study of EEG/MEG signals (LFPy, The Virtual Brain, and Dynamic Causal Modeling of EEG/MEG data), HNN is unique in its model, goals, and capabilities. To highlight these differences, we have re-written and expanded the Discussion section on the topic which is now titled “Comparison to other neocortical models and EEG/MEG modeling software”.

We have also attempted to clarify confusion on the relationship between source localized EEG/MEG signals (primary current dipoles), LFP, and EEG/MEG sensor level data. Please see response to comment #1 above. We emphasize that as currently constructed, HNN is specific to simulating source localized primary current dipoles in a single brain region of interest from the intracellular current flow in pyramidal neuron dendrites. Future directions are aimed at including simulations of LFP, and sensor level EEG/MEG signals in HNN’s vizualization and distribution.

References

1. Hämäläinen M, Hari R, Ilmoniemi R, Knuutila J, and Lounasmaa OV, Magnetoencephalography – theory, instrumentation, and applications to noninvasive studies of the working human brain. Reviews of Modern Physics, 65: 413-497, 1993.

2. Hari R and Ilmoniemi RJ, Cerebral magnetic fields. Crit Rev Biomed Eng, 14: 93-126, 1986.

3. Ikeda, H., Wang, Y., and Okada, Y. C. (2005). Origins of the somatic N20 and high-frequency oscillations evoked by trigeminal stimulation in the piglets. *Clinical Neurophysiology, 116*(4), 827–841. https://doi.org/10.1016/j.clinph.2004.10.010

4. Okada, Y. C., Wu, J., and Kyuhou, S. (1997). Genesis of MEG signals in a mammalian CNS structure. *Electroencephalography and Clinical Neurophysiology, 103*(4), 474–485. https://doi.org/10.1016/S0013-4694(97)00043-6

5. Murakami, S., and Okada, Y. (2006). Contributions of principal neocortical neurons to magnetoencephalography and electroencephalography signals. *Journal of Physiology, 575*(3), 925–936. https://doi.org/10.1113/jphysiol.2006.105379